

# A Bayesian Framework Based on Gaussian Mixture Model and Radial Basis Function Fisher Discriminant Analysis for Flood Spatial Prediction (BayGmmKda V1.1)

Dieu, Tien Bui[1] and Nhat-Duc, Hoang[2]

[1]Geographic Information System Group, Department of Business Administration and Computer Science, University College of Southeast Norway (USN), Hallvard Eikas Plass, N-3800, Bø I Telemark, Norway.

[2]Faculty of Civil Engineering, Institute of Research and Development, Duy Tan University, P809 - K7/25 Quang Trung, Danang, Vietnam.

*Correspondence to*: Nhat-Duc Hoang (hoangnhatduc@dtu.edu.vn)

**Abstract.** In this study, a probabilistic model, named as BayGmmKda, is proposed for flood assessment with a study area in Central Vietnam. The new model is essentially a Bayesian framework constructed a combination of Gaussian Mixture Model, Radial Basis Function Fisher Discriminant Analysis, and a Geographic Information System database. Experiments used for measuring the model performance point out that the hybrid framework is superior to other benchmark models including the adaptive neuro fuzzy inference system and the support vector machine. To facility the model implementation, a
software program of BayGmmKda has been developed in Matlab environment. The newly proposed model is shown to be a useful tool for flood susceptibility evaluation.

**Key words**: Flood Evaluation; Bayesian Classifier; Gaussian Mixture Model; Discriminant Analysis; Latent Variable.

## 1 Introduction

Flood stands out as one of the most destructive phenomena featured by its immense scale of damages as well as its large
spatial extent (Dottori et al., 2016b). Catastrophic flood events destroy crops, infrastructures, and bring out heavy loss of human lives. Floods also lead to siltation of the reservoirs and thus limit the capacity of dams designed for flood control (Sanyal and Lu, 2004). This natural hazard is known to negatively affect 170 million people around the globe annually



(Kazakis et al., 2015;Judi et al., 2011). On average, people in more than 90 countries are victimized by catastrophic flooding, resulting in more than 170,000 deaths (Kazakis et al., 2015).

Because of monsoonal rainfalls and cyclonic patterns, regions in Southern Asian are considerably affected by flood (Loo et al., 2015). Particularly in Vietnam, floods are often triggered by tropical cyclones and this country has undergone destructive consequence of flooding in many provinces. According to Tien Bui et al. (2016b),  more than 70% of the population and roughly 60% of the area in this nation are negatively affected by flood hazards. Based on a report done by Kreft et al. (2014), in the last two decades, flood accounts for destructions that cost approximately $2.9 billion in Vietnam.

In addition, the occurrences of flood in Vietnam are expected to soar in the near future due to the increases of poorly planned infrastructure developments and urbanization near watercourses, as well as an increased activity of deforestation and climate change. Hence, accurate forecasting of flood becomes an increasingly crucial task for land-use planning as well as disaster mitigation strategies. The reason is that it is possible to forecast flood events and flood-prone area can be identified by means of intelligent approaches at a region scale (Kia et al., 2012;Tien Bui et al., 2016b).

Needless to say, the identification of flood susceptible areas can significantly reduce its damages to the national economy and human lives by avoiding infrastructure developments and densely populated settlements in the highly flood susceptible areas (Zhou et al., 2016). The prediction outcomes also help Government agency to issue appropriate flood management policies and to focus its limited financial resource to construct large-scale flood defense infrastructure in areas that possess great economic values but are highly susceptible to flood (Bubeck et al., 2012). Therefore, a framework of

accurate flood prediction is of great usefulness.

Recently, there is an increasing trend of applying Geographic Information System (GIS) in flood prediction. GIS technology is demonstrated to be a helpful tool to investigate the multi-dimensional events of flooding (Tien Bui et al., 2016a;Shahabi and Hashim, 2015;Jia et al., 2016). GIS method is capable of simultaneously analyzing different layers of information (Candy et al., 2014), such as flood inventory map and various flood conditioning factors (e.g. topological and

hydrological features) to yield flood susceptibility evaluation results. In addition, flood forecasting systems based on GIS are indeed suitable for a participatory approach to flood management; the reason is that this technique facilitates the communication with the public in a scientific manner.



To construct flood susceptibility evaluation models, databases of GIS that contains a set of flood influencing factors and information of past flood events is established at the first step. At the next step, advanced soft computing models can be

utilized to distinguish the flood vulnerable areas from the entire studied region (Tehrany et al., 2015b). In this way, the flood prediction problem boils down to a supervised classification task. Nevertheless, most models in the current studies can only yield qualitative outputs of flood prediction outcome (i.e. flood–no flood) (Dottori et al., 2016a); probabilistic evaluations have rarely been seen in the literature. Given these motivations, this study proposes a novel methodology designed for enhancing the prediction accuracy as well as deriving probabilistic evaluation of flood susceptibility in a regional scale.

The method is relied on Bayesian framework with the Gaussian mixture model (GMM) and the Radial Basis Function Fisher Discriminant Analysis (RBFDA). GMM is employed for density approximation to calculate the posterior probability of flood within the Bayesian framework. Furthermore, to boost the classification accuracy of the Bayesian model, RBFDA is employed to construct a latent variable for from the original GIS database; this latent variable aims at maximizing data discrimination with respect to the two classes of 'flood' and 'no-flood'.

In essence, the proposed integrated framework contains two phases of analysis. RBFDA is first employed for latent variable construction. The Bayesian approach assisted by GMM is then used to perform probabilistic pattern recognition. The first level performs pattern discriminant analysis task and the second level carries out the prediction to derive the model output of flood evaluation. Based on previous studies indicating that hierarchical model structure can yielded improving prediction accuracy (Chou and Tsai, 2012;Shahangian and Pourghassem, 2016;Hoang and Tien Bui, 2016), the proposed

unified framework can potentially bring about desirable flood assessment results. The subsequent parts of this study are organized in the following order: Related works on flood prediction are summarized in the second section. The next section introduces the research method of the current paper. The fourth part describes the proposed Bayesian model for flood susceptibility forecasting. The next part reports the model prediction accuracy and comparison. The last section of the paper discusses some conclusions on this work.

**2 A Review of Related Works on Flood Susceptibility Prediction**

Because of the criticality of flood prediction, this problem has gained an increasing attention from the academic community. Following this trend, various flood analyzing tools have been developed, ranging from relatively simple



methods to more sophisticated methodologies involving hydrological and hydraulic models (Winsemius et al., 2013;Papaioannou et al., 2015). In recent years, remote sensing coupled with the advancement of GIS technology has been

increasingly shown to be a reliable method for producing synoptic coverage over a large area in a cost effective way (Sanyal and Lu, 2004;Tien Bui et al., 2016b;Kazakis et al., 2015;Kia et al., 2012).

The new approach based on GIS successfully evades the limitation of the hydrological models and equips decision-makers with a powerful flood analysis tools. GIS databases integrated with data-driven methods have demonstrated their effectiveness and accuracy in large scaled flood predictions. An fuzzy logic based algorithm has been used to develop a map

of flooded areas from synthetic aperture radar imagery, used for the operational flood management system in Italia, was established by Pulvirenti et al. (2011).  A model based on the frequency ratio approach and GIS for spatial prediction of flooded regions was first introduced by Lee et al. (2012); the spatial database were constructed by field surveys and maps of the topography, geology, land cover, and infrastructure.

Prediction models with artificial neural network (ANN) have been employed for flood susceptibility evaluation by

various scholars (Kia et al., 2012;Seckin et al., 2013;Rezaeianzadeh et al., 2014;Radmehr and Araghinejad, 2014); previous works have shown ANN as a capable nonlinear modeling tool. Nevertheless, ANN learning is prone to overfitting and its performance has been shown to be inferior to that of support vector machine (Hoang and Pham, 2016).

Kazakis et al. (2015) introduced a multi-criteria index to assess flood hazard areas that relies on GIS and Analytical Hierarchy Process (AHP); in this methodology, the relative importance of each flood conditioning factors for the occurrence

and severity of flood were determined via AHP. More recently, Support Vector Machine-based flood susceptibility analysis approaches have been proposed by Tehrany et al. (2015a) and Tehrany et al. (2015b); the research finding is that SVM is more accurate than other benchmark models including the decision tree classifier and the conventional frequency ratio model.

Mukerji et al. (2009) constructed flood forecasting models based on an adaptive neuro-fuzzy interference system

(ANFIS), Genetic Algorithm optimized ANFIS, ; experiments demonstrated that ANFIS attained the most desirable accuracy. Recently, a metaheuristic optimized neural fuzzy inference system, named as MONF, has been introduced by Tien



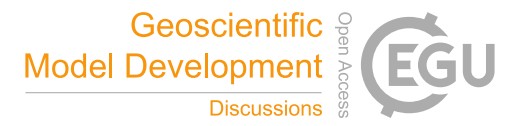

Bui et al. (2016b); the research finding is that MONF is more capable than decision tree, ANN, SVM, and conventional ANFIS.

As can be seen from the literature review, various data-driven and advanced soft computing approaches have been
proposed to construct different flood forecasting models. In most of previous studies, the flood prediction was formulated as a binary pattern recognition problem in which the model output is either flood or no flood. Probabilistic models have rarely been examined to cope with the complexity as well as uncertainty of the problem under concern. Therefore, our research aims at enriching the body of knowledge by proposing a novel Bayesian probabilistic model to estimate the flood vulnerability with the use of a GIS database.

**3 Research Method**

**3.1 Flood inventory map and flood conditioning factors of the study area**

**3.1.1 The study area**

In this research, Tuong Duong district (central Vietnam) is selected as the study area (see **Figure 1**). This is recognizably a heavily affected flood region in the country (Reynaud and Nguyen, 2016). The area of the district is
approximately 2803 km$^2$ and locates between the longitudes of 18°58'42"N and 19°39'16"N, and between the latitudes of 104°15"58'E and 104°55"57'E. The topographical feature of the Tuong Duong district is inherently complex with mountainous areas, watersheds, and rivers. Drastic floods often divided the district into several isolated areas which are very difficult to approach for rescuing or evacuation purposes.

The district is featured by two separated seasons, namely a cold season (from November to March) and a hot season
(from April to October). The yearly rainfall of the district is within the range of 1679 mm and 3259 mm. The rainfall amount is primarily intensified during the rainy period which contributes to roughly 90% of the total annual rainfall. Due to the district's location as well as its topographic and climatic features, the study area is highly susceptible to flood events with immense infliction to human casualty and economic value. An examination carried out by Reynaud and Nguyen (2016) reported that approximately 40% of families have damaged by floods and roughly 20% of families must be departed from
the flooded areas; the average loss of flood goes up to 24% of the family income each year.



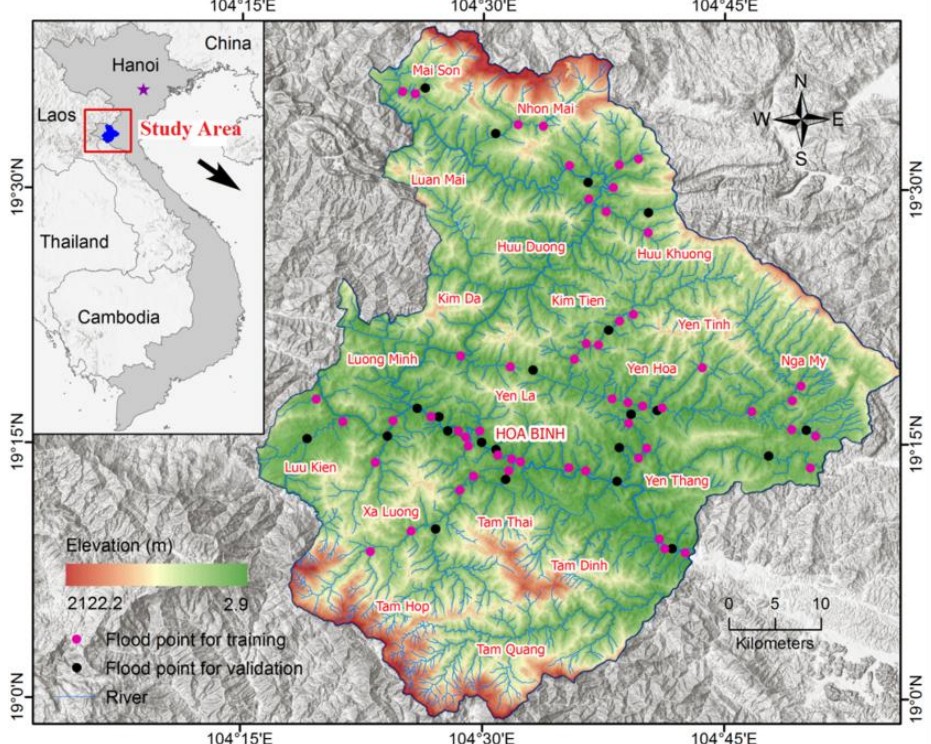

**Figure 1** Tuong Duong district (Central Vietnam)

### 3.1.2 Flood inventory map

Prediction of flood zones can be based on an assumption that future flood events are governed by the very similar

conditions of flooded zones in the past. Therefore, it is a reasonably strategy to analyze past records of flood occurrences

(Tien Bui et al., 2016b;Tehrany et al., 2015b). The first step of this analysis is to establish a flood inventory map for the

region under investigation.

The flood inventory map established in the current research stores documentations of past flood events (see **Figure 1**).

The map was constructed by gathering information of the study area, field works at flood areas, and analyses from results of

the Landsat 8 Operational Land Imagery (from 2010 to 2014) with the resolution of 30m (retrieved from

http://earthexplorer.usgs.gov). Furthermore, the location of flood events was also verified by field works carried out in 2014

with handhold GPS devices. In summary, the total number of flood locations during the last five years was recorded to be 76.

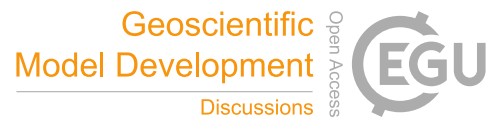

### 3.1.3 Flood influencing factors

In order to construct a flood prediction model, besides the flood inventory map, it is crucial to determine the flood

influencing factors (Tehrany et al., 2015a). It is worth to notice that the selection of the flood governing factors varies due to

different characteristics of study areas and the availability of data (Papaioannou et al., 2015).

**Table 1** Flood influencing variables and their category descriptions

| Factors | Notation | Description of factor categories |
|---|---|---|
| Slope (°) | $IF_1$ | 1 (0 to 0.5); 2 (0.5 to 2); 3 (2 to 5); 4 (5to 8); 5 (8 to 13); 6 (13 to 20); 7 (20 to 30); 8 (>30) |
| Elevation (100m) | $IF_2$ | 1 (<1); 2 (1 to 2); 3 (2 to 3); 4 (3 to 4); 5 (4 to 5); 6 (5 to 6); 7 (6 to 7); 8 (7 to 10); 9 (10 to 13); 10 (>13) |
| Curvature | $IF_3$ | 1 (<-2); 2 (-2 to -0.05) ; 3 (-0.05 to 0.05); 4 (0.05 to 2); 5 (>2) |
| Topographic Wetness Index (TWI) | $IF_4$ | 1 (<6.5); 2 (6.5 to 7.5); 3 (7.5 to 8.5); 4 (8.5 to 9.5); 5 (9.5 to 10.5); 6 (10.5 to 11.5); 7 (11.5 to 12.5); 8 (>12.5) |
| Stream Power Index (SPI) | $IF_5$ | 1 (<1); 2 (1 to 3); 3 (3 to 5); 4 (5 to 7); 5 (7 to10); 6 (10 to 15); 7 (15 to 20); 8 (20 to 30); 9 (30 to 50); 10 (>50) |
| Distance to river (m) | $IF_6$ | 1 (<40); 2 (40 to 80); 3 (80 to 120); 4 (120 to 200); 5 (200 to400); 6 (400 to 700); 7 (700 to 1500); 8 (>1500) |
| Stream density (km/km2) | $IF_7$ | 1 (<1); 2 (1 to 3); 3 (3 to 5); 4 (5 to 7); 5 (7 to9); 6 (>9) |
| Normalized Difference Vegetation Index (NDVI) | $IF_8$ | 1 (<0.3); 2 (0.3to 0.35); 3 (0.35 to 0.4); 4 (0.4 to 0.45); 5 (0.45 to0.5); 6 (0.5 to 0.55); 7 (0.55 to 0.6); 8 (>0.6) |
| Lithology (rock type) | $IF_9$ | 1 (Q); 2 (Nkb); 3 (Jmh); 4 (T3npb); 5 (T2); 6 (C-bslk); 7 (D-ntdl); 8 (S2-D1hn); 9 (O3-S1sc3); 10 (O3-S1sc2); 11 (O3-S1sc1); 12 (PR2bk) |
| Rainfall (1000mm) | $IF_{10}$ | 1 (<1.82); 2 (1.82 to 1.92); 3 (1.92 to 2.02); 4 (2.02 to 2.12); 5 (2.12 to 2.22); 6 (2.22 to 2.32); 7 (2.32 to 2.42); 8 (>2.42) |

In this study, the flood conditioning factors were selected by literature review and field data. Accordingly, ten factors

are chosen to analyzing flood vulnerability and a GIS database consisting of the flood inventory map and the chosen factors

was established. The ten flood governing factors can be listed as: slope ($IF_1$), elevation ($IF_2$), curvature ($IF_3$), topographic

wetness index (TWI) ($IF_4$), stream power index (SPI) ($IF_5$), distance to river ($IF_6$), stream density ($IF_7$), normalized

difference vegetation index (NDVI) ($IF_8$), lithology ($IF_9$), and rainfall ($IF_{10}$). The information of ten conditioning factors of



flood occurrence employed in this study is summarized in **Table 1**. The distributions of the ten factors within the studied

region are illustrated in **Figure 2**.

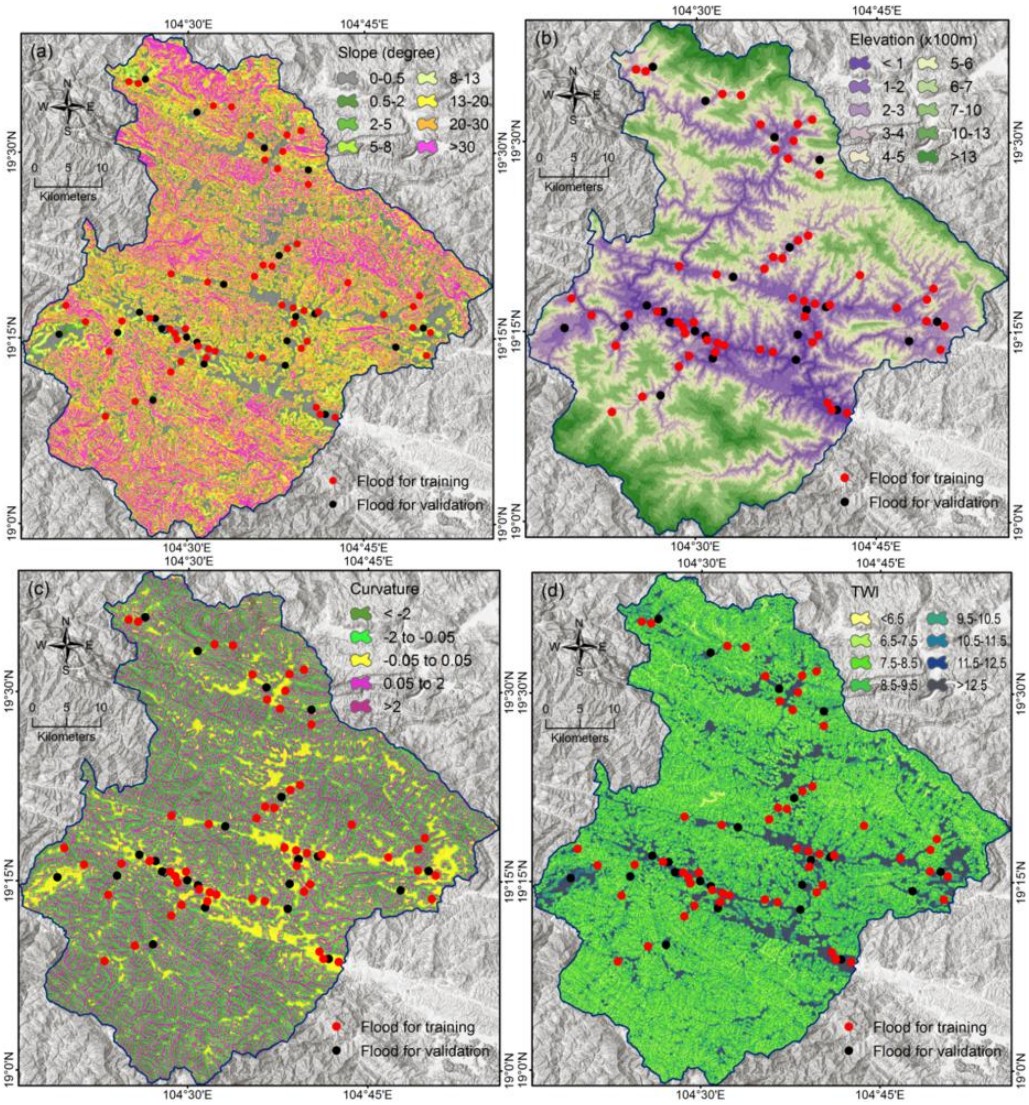

**Figure 2** Flood influencing factors: (a) Slope, (b) Elevation, (c) Curvature, (d) Topographic wetness index






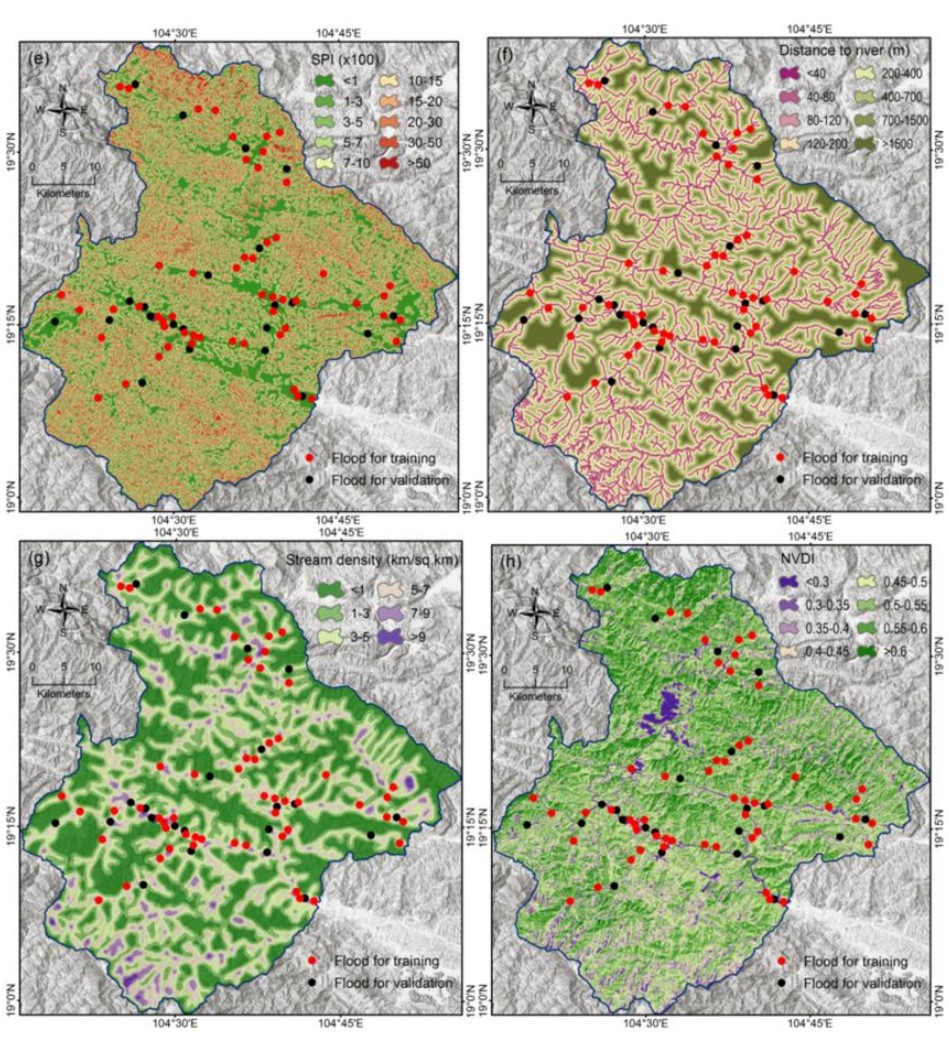

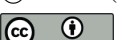


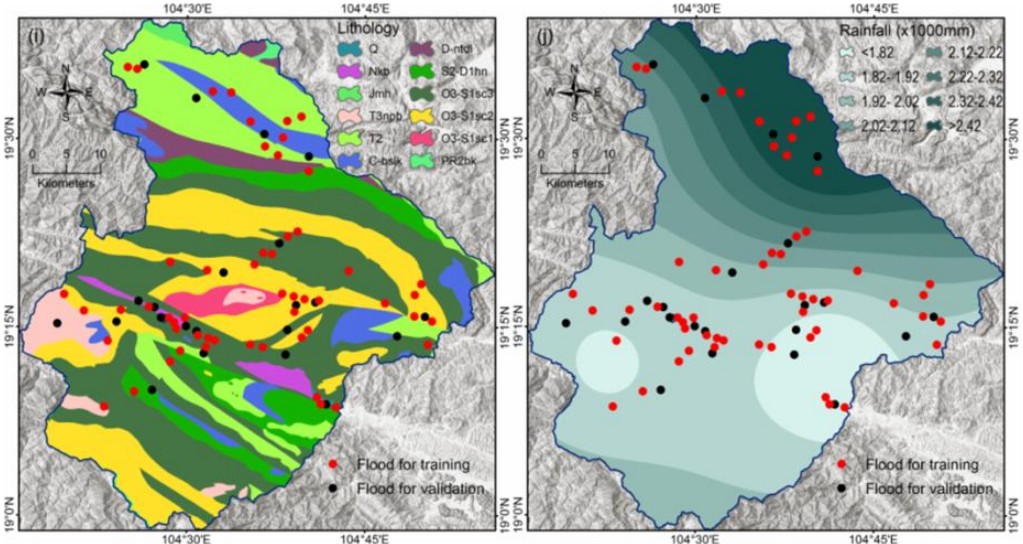

**Figure 2 (Cont.)** Flood influencing factors: (e) Stream power index, (f) Distance to river, (g) Stream density, (h) Normalized

Difference Vegetation Index, (i) Lithology, (j) Rainfall

**3.2 Bayesian Framework for Pattern Classification**

To cope with the complexity as well as the uncertainty of the problem of interest, Bayesian framework is employed in this study to evaluate the flood susceptibility of each data sample. The Bayesian framework provides a flexible way for

probabilistic modeling. This method features a strong ability for dealing with uncertainty and noisy data (Theodoridis, 2015;Cheng and Hoang, 2016). Nevertheless, previous studies have rarely examined the capability of this approach for inferring flood susceptibility.

Basically, pattern classification aims at assigning a pattern to one of $M$ distinctive class labels $C_k$ ($k = 1,…, M$). To recognize an input pattern based on the information supplied by its feature vector $X$, we need to attain the pose probability

$P(C_k|X)$, which indicates the likelihood that the feature vector $X$ falls into a certain group $C_k$. Based on such information, the pattern will be categorized to the group with the highest posterior probability. The posterior probability $P(C_k|X)$ is calculated as follows (Webb and Copsey, 2011):

$$P(C_k \mid X) = \frac{p(X \mid C_k) \times P(C_k)}{p(X)} \qquad (1)$$





where $P(C_k \mid X)$ denotes the posterior probability. $p(X \mid C_k)$ represents the likelihood which is also called the class-

conditional probability density function (PDF). $P(C_k)$ denotes the prior probability, which implies the probability of the

class before any feature is measured. The denominator $p(X)$ is the evidence factor; this quantity is merely a scale factor for

guaranteeing that the posterior probabilities are valid; it can be calculated as follows:

$$P(X) = \sum_{k=1}^{M} p(X \mid C_k) \times P(C_k) \qquad (2)$$

Generally, the prior probabilities $P(C_k)$ can be calculated by computing the ratio of training instances in each class.

Thus, the bulk of establishing a Bayesian classification model is to calculate the likelihood $p(X/C_k)$. This likelihood

expresses the density of input patterns in the learning space within a certain group of data. In most of situations, $p(X/C_k)$ is

unknown and must be estimated from the available data. In this research, the Gaussian mixture model is utilized for

computing the $p(X/C_k)$ quantity.

**3.3 Gaussian Mixture Model for Density Estimation**

**3.3.1 Gaussian Mixture Model (GMM)**

Generally, density estimation can be defined as the problem of approximating a PDF given a finite number of data

instances (Scott, 2015). GMM have been shown to be an effective parametric method for modeling of data distribution

especially in high dimensional space (McLachlan and Peel, 2000;Theodoridis and Koutroumbas, 2009). Previous studies

(Paalanen, 2004;Figueiredo and Jain, 2002;Gómez-Losada et al., 2014;Arellano and Dahyot, 2016) point out that any

continuous distribution can be approximated arbitrarily well by a finite mixture of Gaussian distributions. Due to their

usefulness as a flexible modeling tool, GMMs have received an increasing attention from the academic community (Zhang et

al., 2016;Khanmohammadi and Chou, 2016;Ju and Liu, 2012).

In a *d*-dimensional space the Gaussian PDF is defined mathematically in the following form:

$$N(x \mid \mu) = \frac{1}{(2\pi)^{d/2} \mid \Sigma \mid^{1/2}} \exp\{-\frac{1}{2}(x - \mu)^T \Sigma^{-1}(x - \mu)\} \qquad (3)$$



where ~ denotes the vector of variable mean and $\Sigma$ represents the matrix of covariance; and $_{,,} = \{ \sim, \Sigma \}$ denotes a set of

distribution parameter.

A GMM is, in essence, an aggregation of several multivariate Normal distributions; hence, its PDF for each data sample

is computed as a weighted summation of Gaussian distributions (see **Figure 3**):

$$p(x \mid \Theta) = \sum_{i=1}^{k} \Gamma_i p_i(x \mid _{,,i}) = \sum_{i=1}^{k} \Gamma_i N(x \mid _{,,i}) \qquad (4)$$

where $\Theta = \{ \Gamma_1, \Gamma_2, ..., \Gamma_k, _{,,1}, _{,,2}, ..., _{,,k} \} . \{ \Gamma_1, \Gamma_2, ..., \Gamma_k, \}$ is called the mixing coefficients of $k$ Gaussian components

and $\sum_{i=1}^{k} \Gamma_i = 1$.

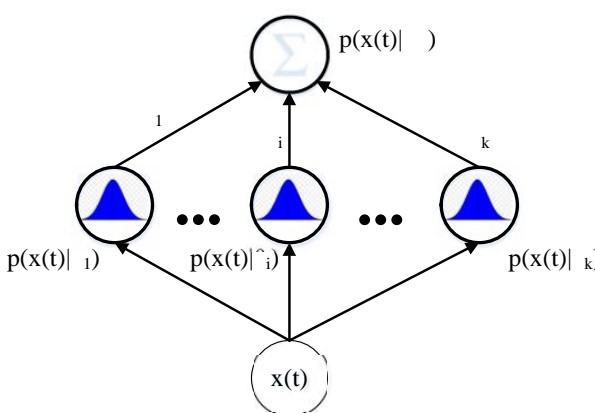

**Figure 3** Structure of Gaussian Mixture Model (GMM)

Accordingly, the PDF for all data samples can be expressed as follows (Ju and Liu, 2012):

$$p(X \mid \Theta) = \prod_{t=1}^{n} p(x_t \mid \Theta) = L(\Theta \mid X) \qquad (5)$$

Identifying a GMM's parameters $\Theta$ can be considered as unsupervised learning task within which a dataset of

independently distributed data points $X = \{ x_1, ..., x_N \}$ generated from an integrated distribution dictated via the PDF

$p(X \mid \Theta)$. The goal is to find the most appropriate value of $\Theta$, denoted as $\Theta_e$, that maximizes the log-likelihood function:





$$\Theta_e = \arg\max_{\Theta} \log(L(X,\Theta)) = \log(\prod_{t=1}^{n} p(x_t|\Theta)) = \sum_{t=1}^{n} \log(\sum_{i=1}^{k} r_i p_i(x_t|_{n\,i})) \qquad (6)$$

Practically, instead of dealing with the log-likelihood function, an equivalent objective function $Q$ is optimized (Ju and

Liu, 2012):

$$Max.\ Q = \sum_{t=1}^{n} \sum_{i=1}^{k} w_{it} \log[r_i p_i(x_t|_{n\,i})] \qquad (7)$$

where $w_{it}$ is a posteriori probability for the $i$th class, $i = 1,...,k$ and $w_{it}$ satisfies the following conditions:

$$w_{it} = \frac{r_i p_i(x_t|_{n\,i})}{\sum_{s=1}^{k} r_s p_s(x_t|_{n\,s})} \ and\ \sum_{i=1}^{k} w_{it} = 1 \qquad (8)$$

The expectation maximization (EM) algorithm is a standard method to compute $\Theta_e$. Besides the EM method, this study

employs an unsupervised learning approach for determining GMMs proposed by Figueiredo and Jain (2002). These two

algorithms are briefly reviewed in the next section of the paper.

### 3.3.2 Learning of finite Mixture Model with the Expectation Maximization (EM) Algorithm

The EM method is a statistical approach to fit a GMM based on historical data; this method converges to a maximum

likelihood estimate of model parameters (McLachlan and Krishnan 2008). It can be recapitulated as follows (McLachlan and

Peel, 2000). Commencing from an initial parameter $\Theta_o$, an iteration of the EM algorithm consists of the *E-step* in which the

current conditional probabilities $p_i(x_t|_{n\,i}) = N(x_i|\sim_i, \Sigma_i)$ that $x_t$ generated from the $i$th mixture component are

calculated, and the *M-step* within which the maximum likelihood estimates of $_{n\,i}$ are updated. The iteration of EM algorithm

terminates when the change value of the objective function is lower than a threshold value.

These two steps of the EM procedure are stated as follows:

*E-step*: estimating the expected classes of all data samples for each class $w_{it}$ based on Eq. (8).

*M-step*: calculating maximum likelihood given the data's class membership distribution using the following equations:



$$\Gamma_i^{new} = \frac{1}{n}\sum_{t=1}^{n} w_{it} \tag{9}$$


$$\tilde{}_i^{new} = \frac{\sum_{t=1}^{n} w_{it} x_t}{\sum_{t=1}^{n} w_{it}} \tag{10}$$

$$\Sigma_i^{new} = \frac{\sum_{t=1}^{n} w_{it}(x_t - \tilde{}_i^{new})(x_t - \tilde{}_i^{new})^T}{\sum_{t=1}^{n} w_{it}} \tag{11}$$

### 3.3.3 Unsupervised learning of finite mixture model

The EM algorithm increases the log-likelihood iteratively until convergence is detected; and this approach generally can derive a good set of estimated parameters. Nonetheless, EM suffers from low convergence speed in some data sets, high

sensitivity to initialization condition, and sub-optimal estimated solutions (Biernacki et al., 2003). Moreover, additional efforts are required to determine an appropriate number of Gaussian distributions within the mixture.

As an attempt to alleviate such drawbacks of EM, Figueiredo and Jain (2002) put forward an unsupervised algorithm for learning a GMM from multivariate data. The algorithm features the capability of identifying a suitable number of Gaussian components autonomously; and by experiments, the authors show that the algorithm is not sensitive to

initialization. In other words, this unsupervised approach incorporates the tasks of model estimation and model selection in a unified algorithm. Generally, this method can initiate with a large number of components. The initial values for component means can be assigned to all data points in the training set; in an extreme case, it is possible to distribute the component number equal to the data point number. This algorithm gradually fine-tunes the number of mixture components by casting out element of Normal distributions that are irrelevant for the data modeling process (Paalanen, 2004).

Furthermore, Figueiredo and Jain (2002) employed the Minimum Message Length (MML) criterion (Wallace and Dowe, 1999) as an index for model selection; the application of this criterion for the case of GMM learning leads to the following objective function (Figueiredo and Jain, 2002):



$$\Omega(\Theta \mid X) = \frac{N}{2} \sum_{i:\Gamma_i > 0} \ln(\frac{n\Gamma_i}{12}) + \frac{C_{nz}}{2}\ln(\frac{n}{12}) + \frac{C_{nz}(N+1)}{2} - \ln L(X,\Theta) \qquad (12)$$

where $n$ denotes the size of the training set, $N$ represents the number of hyper-parameters needed to construct a Gaussian

distribution, and $C_{nz}$ is the number of Gaussian distribution component featuring nonzero weight ($\Gamma_i > 0$). Accordingly,

the EM method is then utilized to minimized Eq. 12 with a fixed number of $C_{nz}$.

In detail, the EM algorithm is employed to estimate $\Gamma_i$ as follows:

$$\Gamma_i^{new} = \frac{\max\{0, (\sum_{t=1}^{n} w_{it}) - \frac{N}{2}\}}{\sum_{j=1}^{k} \max\{0, (\sum_{t=1}^{n} w_{jt}) - \frac{N}{2}\}} \qquad (13)$$

Accordingly, the parameters $\sim_i^{new}$ and $\Sigma_i^{new}$ are updated based on Eq. 10 and 11, respectively. The algorithm stops

when the relative decrease in the objective function $\Omega(\Theta \mid X)$ becomes smaller than a preset threshold (e.g. $10^{-5}$).

**3.4 Radial Basis Function Fisher Discriminant Analysis for Latent Variable Generation**

In machine learning, discriminant analysis presents a highly useful technique to construct relevant input patterns from

the original data set. This technique aims at unraveling the underlying structure of the data which is helpful for pattern

recognition. Introduced by Mika et al. (1999), the Radial Basis Function Fisher Discriminant Analysis (RBFDA) is an

extension of the Fisher Discriminant Analysis for dealing with data nonlinearity. RBFDA can be conveniently utilized to

project the feature from the original learning space to a projected space that expresses a high degree of class separability

(Theodoridis and Koutroumbas, 2009). Using this kernel technique, the data from an input space $I$ is first mapped into a high

dimensional feature space $F$. Hence, discriminant analysis tasks can be performed nonlinearly in $I$.

Herein, $W(.)$ is defined as a transformation from an input space $I$ to a high dimensional feature space $F$, to compute $w$

(the projecting vector), it is necessary to maximize the Fisher discriminant ratio as follows:

$$J(w) = \frac{w^T S_B^w w}{w^T S_W^w w} \qquad (14)$$





where $\quad S_B^{\text{w}} = (m_1^{\text{w}} - m_2^{\text{w}})(m_1^{\text{w}} - m_2^{\text{w}})^T$ (15)

$$S_W^{\text{w}} = \sum_{k=1}^{C} \sum_{i=1}^{Nk} (\text{W}(x_i) - m_k^{\text{w}})(\text{W}(x_i) - m_k^{\text{w}})^T$$ (16)

$$m_k^{\text{w}} = \frac{1}{Nk} \sum_{i=1}^{Nk} \text{W}(x_i^k)$$ (17)

To obtain $w$, the kernel trick is applied. Thus, one only needs to establish a formulation of the algorithm which only

requires dot-product $\text{W}(x).\text{W}(y)$ of the training data and employ kernel functions which calculate $\text{W}(x).\text{W}(y)$. The widely-

employed Radial Basis Kernel Function (RBKF) is expressed in the following formula (with    denotes the kernel function

bandwidth):

$$K(x,y) = \exp\left(-\frac{\|x-y\|^2}{2\dagger^2}\right)$$ (18)

Since a solution of the vector $w$ lies in the span of all data samples in the projected space, the transformation vector $w$

is shown in the following formula:

$$w = \sum_{i=1}^{N} \Gamma_i \text{W}(x_i)$$ (19)

From Eq. (17) and Eq. (19), we have: $\quad w^T m_k^{\text{w}} = \frac{1}{Nk} \sum_{j=1}^{N} \sum_{i=1}^{Nk} \Gamma_j k(x_j, x_i^k) = \Gamma^T M_k$ (20)

where $\quad M_k = \frac{1}{Nk} \sum_{i=1}^{Nk} k(x_j, x_i^k)$

Taking into account the formulas of $J(w)$, $S_B^{\text{w}}$, as well as Eq. (20), we can restate the numerator of Eq. (14) in the

following manner:

$$w^T S_B^{\text{w}} w = \Gamma^T M \Gamma$$ (21)

where $M = (M_1\text{-}M_2)(M_1\text{-}M_2)^T$





Similarly, based on the Eq. (17) that defines $m_k^w$, the denominator of Eq. (14) can be demonstrated in the following

way:

$$w^T S_W^w w = \digamma^T N \digamma \qquad (22)$$

where $N = \sum_{k=1}^{2} K_k (I - 1_{l_k}) K_k^T$ ; $K_k$ denotes a $N$-by-$N_k$ kernel matrix with a typical element is $k(x_n, x_m^k)$ . $I$ represents the

identity matrix and $1_{l_k}$ is a matrix within which all positions are $1/l_k$ .

Considering all Eq. (14), Eq. (21), and Eq. (22), the solution of RBFDA can be found by maximizing:

$$J(\digamma) = \frac{\digamma^T M \digamma}{\digamma^T N \digamma} \qquad (23)$$

The optimization problem with the objective function expressed in Eq. (23) is found by identifying the primal

eigenvector of $N^1 M$. Based on the optimization results, an input patter in $I$ is projected on to a line defined by the vector $w$ in

the following manner:

$$w.W(x) = \sum_{i=1}^{N} \digamma_i k(x_i, x) \qquad (24)$$

**4 The proposed Bayesian Framework for Flood Susceptibility Prediction**

**4.1 The established GIS database**

To formulate a flood assessment model, the first stage is to construct a GIS database (see **Figure 4**) within which

locations of past flood events, maps of topographic feature, Landsat 8 data, maps of geological feature, and precipitation

statistical records are acquired and integrated. In this study, the data acquisition, processing, and integration were performed

with ArcGIS (version 10.2) and IDRISI Selva (version 17.01) software packages.

Furthermore, a C++ application has been developed by the authors to transform the flood susceptibility indices into a

GIS format for ArcGIS implementation. Accordingly, the compiled outcomes are employed to form a database that includes

the aforementioned flood conditioning features with two class outputs: "flood" and "no-flood". As mentioned earlier, a total

of 76 flood locations has been recorded. To balance the dataset and reliably construct the flood prediction model, 76



locations of non-flood areas are randomly sampled and included for analysis. Hence, the total database consists of 152 data

samples.

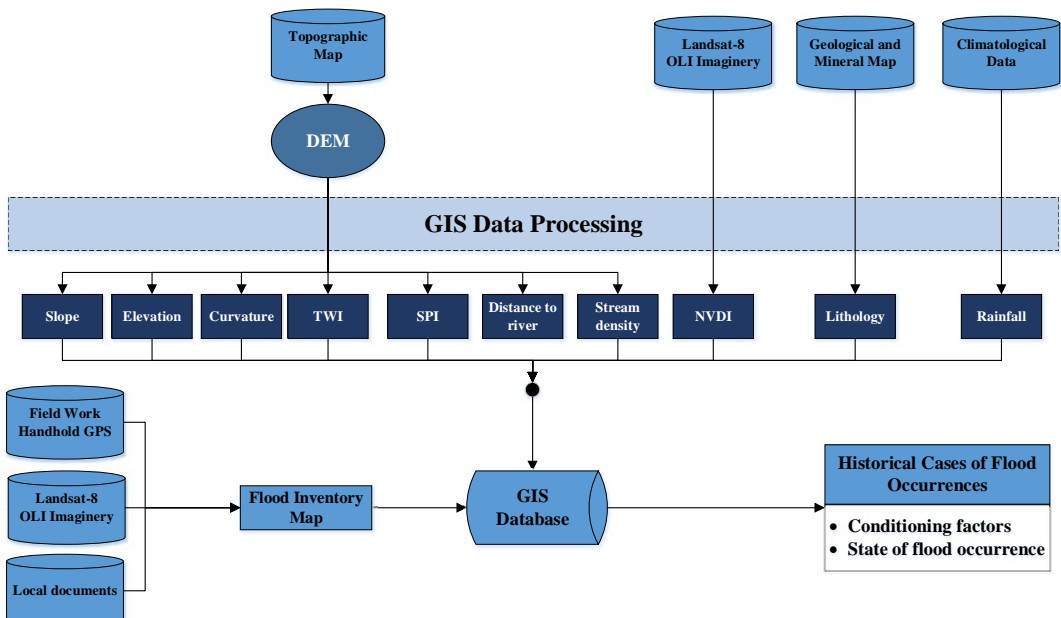

**Figure 4** The established GIS database


### 4.2 The Proposed Model Structure

The proposed model for flood vulnerability assessment that incorporates RBFDA, the Bayesian classification

framework, and GMM is presented in this section of the study. As stated earlier, the model employs RBFDA as the first level

of analysis to generate a latent input factor. This RBFDA-based latent factor aims at expressing a data projection that

features the highest data separation. The Bayesian classification framework coupled with GMM is utilized in the next level

of analysis within which this framework analyzes all relevant conditioning variables and the newly created latent variable to

derive the evaluation result. The overall flowchart of the proposed Bayesian framework based on GMM and RBFDA for

flood susceptibility prediction, named as BayGmmKda, is demonstrated in **Figure 5**.





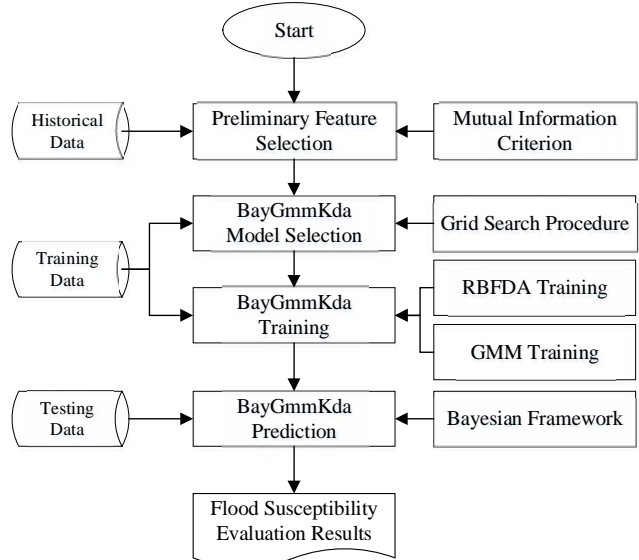


**Figure 5** The proposed BayGmmKda

To construct and evaluate the prediction model, the whole dataset, including 152 data samples, are separated into two

sets: Training Set (90% or 137 samples) employed for model establishing and Testing Set (10% or 15 samples) used for

model testing. It is noted that the input variables of the dataset has been normalized using the Min-Max normalization; the

purpose of data normalization is to hedge against the situation of unbalanced variable magnitudes.

Furthermore, it is beneficial to equip the model with an initial feature selection procedure in which the relevancies of

flood influencing factors are examined. In this research, the mutual information (Kwak and Choi, 2002;Hoang et al., 2016), a

widely employed criterion for feature selection in machine learning, is selected to express the pertinence of each influencing

factor to the flood evaluation outcomes (either flood or no-flood). Basically, the mutual information can be defined as a

measure of the mutual dependence between the two random variables; this criterion quantifies the amount of information

that can be attained about one random variable through the information of another one (Qian and Shu, 2015). It is noticed

that the larger the mutual information, the stronger the relevancy between the flood conditioning factor and the class output.

In addition, to establish the BayGmmKda model, it is required to provide the hyper-parameters of the quantity of

mixture components ($k$) used in GMM and the kernel function bandwidth ($†$) used in RBFDA. It is worth reminding that to



train GMM for density estimation, the model employs two methods: the EM algorithm and the unsupervised algorithm. In case of the EM algorithm, this study employs the Akaike information criterion (AIC) (Akaike, 1974) to identify an appropriate the number of $k$. The value of $k$ is allowed to vary from 1 to 20; AIC is then used to select a model that exhibits a good fit to the data and concurrently requires a few number of mixture components, which indicates less complexity (Olivier

et al., 1999). When the unsupervised GMM learning (Figueiredo and Jain, 2002) is used, the model starts with a maximum component number of 20, the algorithm autonomously carries out the model selection process by removing irrelevant mixture components.

On the other hand, a simple grid search procedure is performed to select a suitable value of the kernel function bandwidth ($\dagger$) used in RBFDA. Within this grid search, the available values of $\dagger$ can be one of the following set:

$\{0.01, 0.05, 0.1, 0.5, 1, 5, 10, 50, 100\}$; in addition, the Training set is subdivided into Training Subset 1 (90%) and Training Subset 2 (10%). Thus, the Training Subset 1 plays the role as a training set for constructing BayGmmKda with a value of $\dagger$; the appropriateness of a hyper-parameter is expressed by the rate of correct classification with the Training Subset 2. The parameter $\dagger$ corresponding to the highest classification accuracy rate is selected for prediction phase.

When a suitable set of tuning parameters, including the number of mixture component $k$ and the kernel function

bandwidth $\dagger$), are properly specified, the training phases of RBFDA and GMM can be carried out. Based on the whole Training Set, RBFDA construct a discriminant analysis based latent variable. GMM is trained based on the original input factors and the RBFDA-based latent factor. Consequently, based on the class conditioning likelihood estimated by GMM, the Bayesian classification framework is formulated to derive a class output (either flood or no-flood) to a novel input pattern.

**4.3 The Developed Matlab Interface of BayGmmKda**

It is noted that GMM with the EM training algorithm is implemented with the Matlab statistical toolbox (MathWorks, 2012b); meanwhile, the BayGmmKda performs the unsupervised algorithm with the program code provided by Figueiredo (2002). The RBFDA algorithm and the unified BayGmmKda model have been coded in Matlab by the authors. In addition, a software program with a graphical user interface (GUI) (see **Figure 6**) for BayGmmKda model implementation has been



coded in Matlab environment by the authors. The GUI development aims at providing a user-friendly system for performing

flood susceptibility predictions.

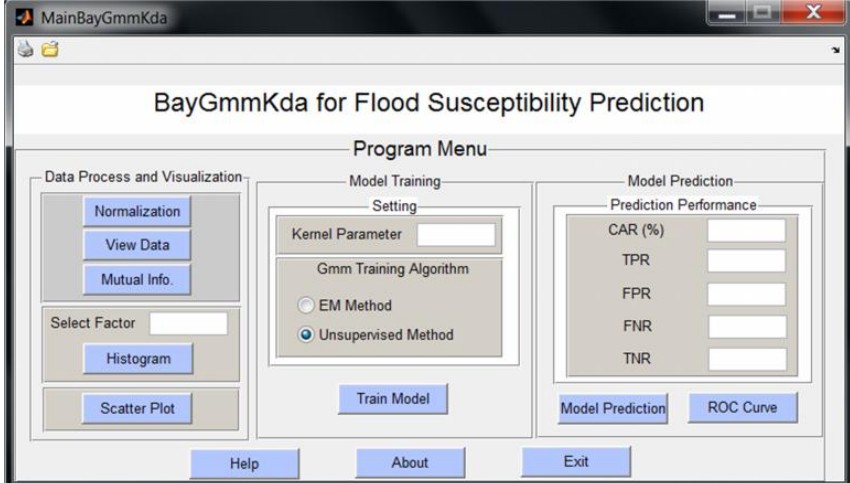

**Figure 6** Main Menu of BayGmmKda

     As shown in **Figure 6, t**he program consists of three modules: Data Process and Visualization, Model Training, and

Model Prediction. The first module provides basic functions for data inspection and visualization including data

normalization, data viewing, and preliminary feature selection with mutual information. In the second module, the users

simply provide model parameters including the kernel function parameter and the GMM training method. The trained model

is employed to carry out prediction tasks in the third module, within which the model prediction performance is reported.

**5. Experimental Results**

**5.1 BayGmmKda Implementation**

     The outcome of the preliminary examination on the pertinence of flood influencing factors is reported in **Figure 7**. As

mentioned earlier, the relevancies of influencing factors are exhibited by the mutual information criterion. Based on the

outcome, $IF_5$ (SPI) features the highest mutual dependence, followed by $IF_7$ (stream density) and $IF_8$ (NVDI). Influencing

factors of $IF_4$ (TWI) and $IF_{10}$ (rainfall) exhibit comparatively low values of mutual information. Because all of the mutual

information values are not null, all conditioning factors deem to be relevant and should be retained for the subsequent

processes of model training and prediction.



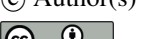

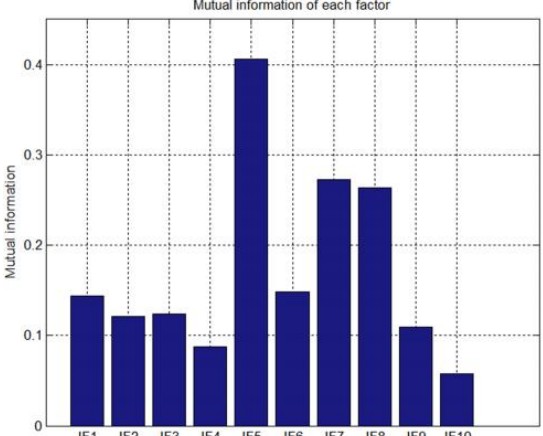

**Figure 7** Mutual information of flood conditioning factors

375        It is worth reminding that the BayGmmKda's training phase are executed in two consecutive steps: RBFDA training and GMM training. RBFDA analyzes the data in Training Set to establishes a latent factor which is a one-dimensional representation of the original input pattern. **Figure 8** illustrates a typical latent factor constructed by RBFDA. In the next step of the training phase, GMM is constructed by the original input patterns with their corresponding labels which consists of ten input factors and with the RBFDA -based latent feature.

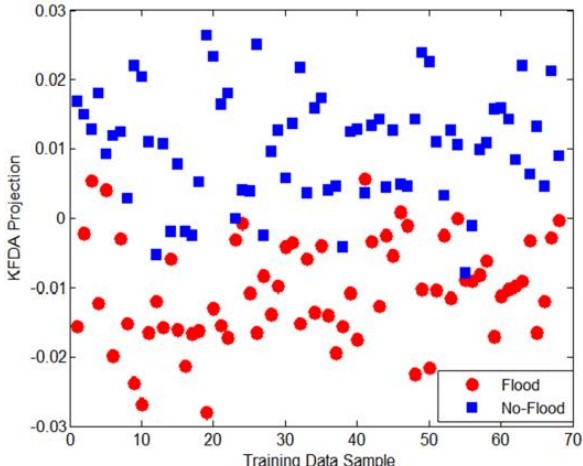


**Figure 8** RBFDA-Based Latent Factor Construction



Commonly, the classification accuracy rate, denoted as CAR, is employed to exhibit the rate of correctly classified instances. In addition, a more detailed analysis on the model capability can be presented by calculating true positive rate TPR, false positive rate FPR, false negative rate FNR, and true negative rate TNR are also commonly utilized to exhibit the

predictive capability of a prediction model (Hoang and Tien-Bui, 2016). The four metrics are calculated as follows:

$$TPR = \frac{TP}{TP + FN} \ ; FPR = \frac{FP}{FP + TN} \ ; \ FNR = \frac{FN}{TP + FN} \ ; \ TNR = \frac{TN}{TN + FP} \tag{25}$$

where $TP$, $TN$, $FP$, and $FN$ represents the values of true positive, true negative, false positive, and false negative, respectively.

Besides, the two indices of TPR and FPR can be graphically summarized by means of Receiver Operating Characteristic

(ROC) curve (van Erkel and Pattynama, 1998). The ROC curve basically demonstrates the trade-off between the two aforementioned TPR and FPR when the threshold for accepting the positive class of 'flood' varies. In addition, the area under the curve of ROC, or AUC for short, can be employed to quantify the classification performance; generally, a better model is characterized by a larger value of AUC.

As aforementioned, the data set are randomly separated into the Training Set and the Testing Set which occupy 90% and

10% of the data samples, respectively. The Training Set is employed to train the mode; meanwhile, the Testing Set is used for validating the model capability after being trained. Since one selection of data for the Training Set and the Testing Set may not truly demonstrate the model's predictive capability, this study carries out a repetitive sub-sampling procedure within which 30 experimental runs is carried out. In each experimental run, 10% of the data set is retrieved in a random manner from the database to constitute Testing Set; the rest of the database are included in the Training Set.

The testing performance of the proposed Bayesian framework for flood susceptibility is reported in **Table 2** and **Figure 9**, which provides the average ROC curves of the proposed model framework, obtained from the random subsampling process, with two methods of GMM training. Herein, the two Bayesian models that employ the EM algorithm and the unsupervised learning algorithms for training GMM are denoted as BayGmmKda-EM and BayGmmKda-UL, respectively. As can be seen from this table, BayGmmKda with the unsupervised learning algorithm demonstrates clearly better predictive

performance (CAR = 89.58%, AUC = 0.94, TPR = 0.96, TNR = 0.91) than that of the BayGmmKda with EM algorithm



(CAR = 86.67%, AUC = 0.93, TPR = 0.95, TNR = 0.85). The performances of BayGmmKda-EM and BayGmmKda-UL are comparable in true positive rates. However, BayGmmKda-UL deems more accurate than BayGmmKda-EM when the two models predicts patterns with negative class label (no-flood).

**Table 2** Prediction Results of BayGmmKda

| Data Set | CAR (%) | AUC | TPR | FPR | FNR | TNR |
|---|---|---|---|---|---|---|
| *Average* | | | | | | |
| BayGmmKda-EM | 86.67 | 0.93 | 0.95 | 0.12 | 0.15 | 0.85 |
| BayGmmKda-UL | 89.58 | 0.94 | 0.96 | 0.12 | 0.09 | 0.91 |
| *Standard deviation* | | | | | | |
| BayGmmKda-EM | 6.51 | 0.07 | 0.05 | 0.10 | 0.12 | 0.12 |
| BayGmmKda-UL | 7.22 | 0.05 | 0.04 | 0.11 | 0.10 | 0.10 |


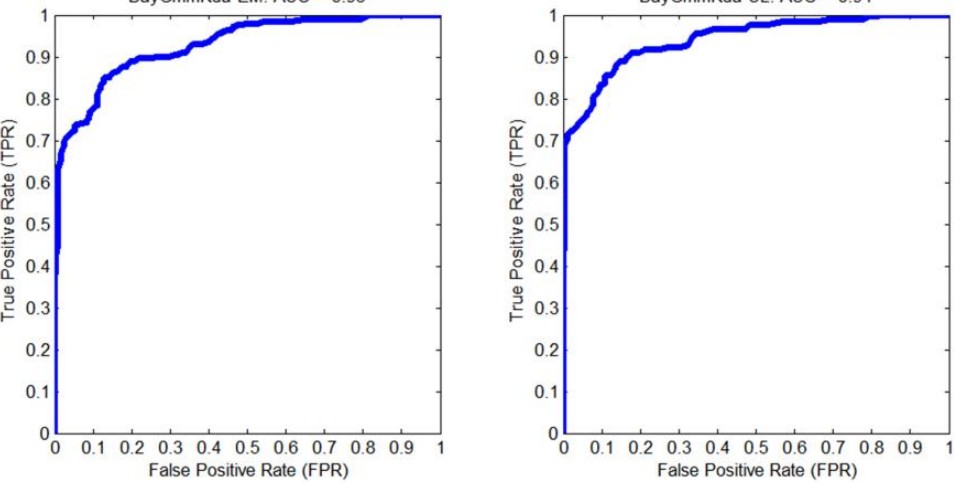

**Figure 9** ROC plots of the proposed BayGmmKda

**5.2 Result Comparison**

In this experiment of the current research, the result of the BayGmmKda model is benchmarked with those of other machine learning models including the support vector machine (SVM), the adaptive neuro fuzzy inference system (ANFIS), and the GMM-based Bayesian Classifier. The above soft computing models are chosen for result comparison because SVM and ANFIS have been recently verified to be effective tools for predicting flood vulnerability (Tien Bui et al.,






2016b;Tehrany et al., 2015b). It is noted that the GMM-based Bayesian Classifier, denoted as BayGmm for short, is the Bayesian framework for classification which employs GMM for density estimation, but is not integrated with the RBFDA algorithm.

Furthermore, comparison between BayGmmKda and BayGmm is helpful to confirm the advantage of the newly constructed BayGmmKda and to verify the usefulness of RBFDA in enhancing the discriminative capability of the hybrid framework. Furthermore, since the performance of BayGmmKda-UL is better than that obtained from BayGmmKda-EM, the proposed BayGmmKda as well as the BayGmm trained by the unsupervised approach (Figueiredo and Jain, 2002) for GMM learning is selected for accuracy comparison in this section.

To construct the SVM model, the model's hyperparameters of the regularization constant ($C$) and the parameter of the radial basis kernel function ($\dagger$) need to be specified. Herein, a grid search process, that is identical to the one used to identify the kernel function bandwidth used in RBFDA, is employed to fine-tuned such hyperparameters of the SVM. It is noted that SVM method is implemented in Matlab package (MathWorks, 2012a). Meanwhile, the ANFIS model used in this section is trained with the metaheuristic approach described in the previous work of Tien Bui et al. (2016b).

**Table 3** Prediction Result Comparison

| Models | CAR (%) | AUC | TPR | FPR | FNR | TNR |
|---|---|---|---|---|---|---|
| *Average* | | | | | | |
| BayGmmKda | 89.58 | 0.94 | 0.96 | 0.12 | 0.09 | 0.91 |
| ANFIS | 85.63 | 0.83 | 0.84 | 0.13 | 0.16 | 0.87 |
| BayGmm | 85.02 | 0.92 | 0.82 | 0.13 | 0.17 | 0.88 |
| SVM | 83.75 | 0.82 | 0.78 | 0.10 | 0.22 | 0.90 |
| *Standard deviation* | | | | | | |
| BayGmmKda | 7.22 | 0.05 | 0.04 | 0.11 | 0.10 | 0.10 |
| ANFIS | 6.17 | 0.05 | 0.14 | 0.10 | 0.14 | 0.10 |
| BayGmm | 7.24 | 0.08 | 0.11 | 0.10 | 0.11 | 0.10 |
| SVM | 10.33 | 0.06 | 0.16 | 0.11 | 0.16 | 0.11 |

It is noted that a random subsampling with 30 runs is employed for all models in this experiment. The result comparison between the proposed BayGmmKda and other benchmark methods is reported in **Table 3**. The experimental outcome shows that out that the proposed BayGmmKda, which is a hybridization of GMM, RBFDA and Bayesian framework, yields the



best value of CAR (89.58%, AUC = 0.94). ANFIS (CAR = 85.63%, AUC = 0.83) is the second best model, followed by BayGmm (85.02%, AUC = 0.92) and SVM (83.75%, AUC = 0.82).

Furthermore, to better confirm the superiority of the proposed method, the Wilcoxon signed-rank test, a non-parametric test, is employed to analyze the model prediction outcomes. This approach is commonly employed to evaluate whether classification outcomes of prediction models are significantly dissimilar. It is noted that the assumption of normality in the data is not required for the implementation of the Wilcoxon signed-rank test. It is noted that the significance level of the statistical test is usually set to be 0.05.

Using this test, the *p*-values st obtained from experimental results of the four models have can be computed; based on the threshold value of 0.05, if the *p*-value falls below the threshold, we can conclude that the prediction outcomes attained from the two models are significantly dissimilar. Outcomes of the Wilcoxon signed-rank test is reported in **Table 4**. It is noted that the signs "++", "+", "--", and "-" represents a significant win, a win, a significant loss, and a loss, respectively. Thus, the test reliably confirms that proposed BayGmmKda achieves significant wins over all other models (ANFIS, BayGmm, and SVM).

**Table 4** Model Comparison Based on the Wilcoxon signed-rank test

|  | BayGmmKda | ANFIS | BayGmm | SVM |
|---|---|---|---|---|
| BayGmmKda |  | ++ | ++ | ++ |
| ANFIS | -- |  | + | + |
| BayGmm | -- | - |  | + |
| SVM | -- | - | - |  |

**5.3 Construction of Flood Susceptibility Map with the Proposed BayGmmKda Model**

Experimental outcomes have indicated that BayGmmKda can deliver the best flood susceptibility prediction for the district of Tuong Duong (in Central Vietnam). Accordingly, in this section, the proposed model is utilized to compute the flood vulnerability index for all pixels within this studied district. The computed outcomes have been converted to GIS format and used to construct a flood susceptibility map by means of the ArcGIS 10.2 software package.

The flood susceptibility map (see **Figure 10**) has been combined with the map of flood inventory to compute the percentage of the flood locations and the percentage of the susceptibility map; accordingly, we can obtain a graphic curve



which visualizes five flood hazard levels: very high (10%), high (10%), moderate (10%), low (20%), and very low (50%). A

further examination on the constructed flood vulnerability map reveals that that 10% of the Tuong Duong district was

categorized as 'very high' and this category covers 73.68% of the total number of flood locations. Meanwhile, the levels of

'high' and 'moderate' both cover 10% of the region and account for 15.79% and 7.9% of the flooded locations, respectively.

The levels of 'low' and 'very low' aggregately cover 70% of the map but they only contain 2.63% of the total number of

flood locations. Particularly, 50% of the district, which is categorized to the hazard level of 'very low', contains no flood

location.  Furthermore, it can be seen that approximately 89% of the recorded floods occurred in the areas of 'very high' and

'high' levels of flood hazard as indicated by BayGmmKda. This fact strongly verifies the reliability of the proposed

Bayesian framework.

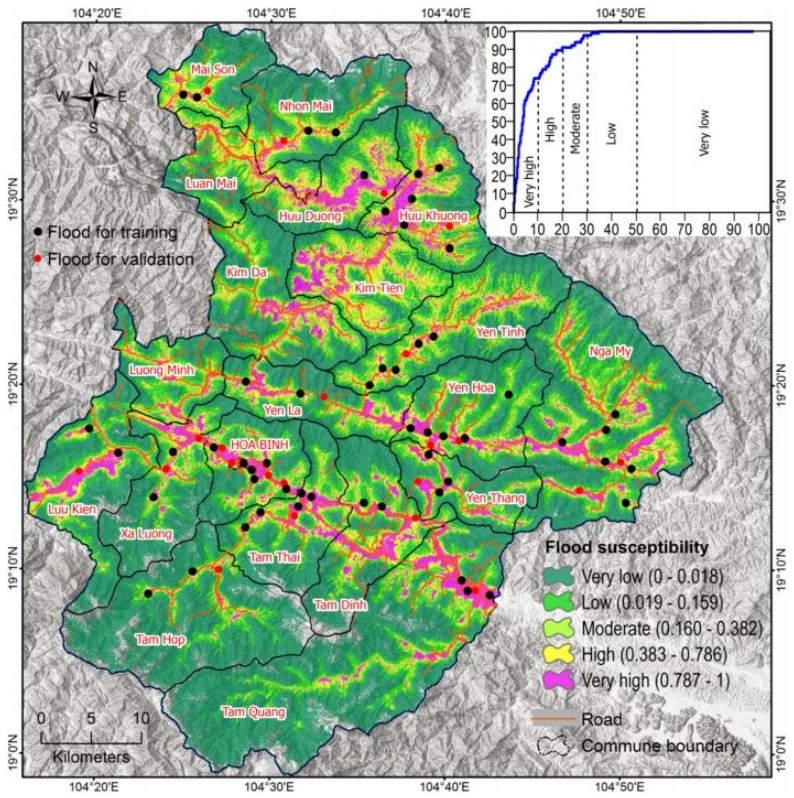

**Figure 10** The constructed flood susceptibility map for the study area



## 6. Conclusion

This research has developed a new tool for flood susceptibility, named as BayGmmKda, evaluation with data collected in the Tuong Duong district (in Central Vietnam). The newly constructed model is a Bayesian framework for classification with an integration of GMM for density approximation and RBFDA for discriminant analysis. A GIS database has been formulated to train and test the BayGmmKda method. The training phase of BayGmmKda can be break down into two steps: discriminant analysis with RBFDA in which a latent factor is generated and density estimation using GMM. After the

training phase, the Bayesian framework is employed to compute the posterior probability of the two class labels (flood and no-flood). Furthermore, a Matlab program with GUI has been developed to ease the implementation of the BayGmmKda model in flood vulnerability assessment.

It is noted that in this study, the GMM training is performed with two methods: the EM algorithm and the unsupervised learning approach. Furthermore, a repeated subsampling process with 30 experimental runs is carried to evaluate the model

prediction outcome. The subsampling process verified by statistical test confirms that the GMM method trained by the unsupervised learning approach has attained a better prediction accuracy compared with the EM algorithm. Therefore, this method of GMM learning is strongly recommended for other studies in the same field.

Furthermore, the experiments demonstrate that the latent factor created by RBFDA is really helpful in boosting the classification accuracy of the BayGmmKda model. This melioration in accuracy of the BayGmmKda stems from its

integrated learning structure. As described earlier, the classification task is performed by a hybridization of discrimination analysis and Bayesian framework. The Bayesian model carried out the classification task by consideration the patterns in the original dataset and an additional factor produced from the discrimination analysis.

Result comparison pointed out that the BayGmmKda is superior to other benchmark approaches including ANFIS and SVM. Therefore, the proposed model, featured by high accuracy and the capability of delivering probabilistic outputs, is a

promising alternative for flood susceptibility prediction. Future extensions of this research may include the model application in flood prediction for other study areas, investigations of other flood conditioning factors which may be relevant for flood analysis, and improving the current model with other novel soft computing methods (e.g. feature selection, pattern classification, dimension reduction, metaheuristic optimization, etc.).



## 7. Code availability

The Matlab code of the BayGmmKda model is given in the Supplement.

## 8. Data availability

The dataset used in this research is given in the Supplement.

## Acknowledgements

Data for this research are from the Project No. B2014-02-21 and were provided by Dr. Quoc-Phi Nguyen (Hanoi University

of Mining and Geology, Vietnam)

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
