# Peer review of "V1.1) for Spatial Prediction of Flood"

_Geoscientific Model Development, 2016_

## Referee Comment (RC1) · Anonymous Referee #1 · 15 Feb 2017

The authors use a data-driven tool to establish a relationship between 10 "flood influencing factors" and the flood itself in a district in Vietnam. While there is a merit in experimenting with statistical tools for trying understand flood occurrence, the current version of the manuscript lacks to demonstrate such a value in using data-driven algorithms for flood prediction. particularly, I have concern with the description and evaluation of the tool. Unless a substantial improvement is made both in the methodology framing and presentation of the manuscript, I would not decremented for publication on GMD.

[Figure]

Major comments: 1) There is no clear objective of the work. And it is not clear how the "tool" can be used for flood mapping or prediction(?). A more focused and talored decription of the tool would be helpful to understand and potentially use for the readers of GMD.

2) The abstract is too short and lacks details of what they attempt to do.

3) There is no definition of flood/no flood. In fact it is not clear at all as to what flood is made in the paper. I think flood extent maps should be used for the evaluation, instead of just the selected points. As it is currently used, then streamflow should be used for the evaluation.

4) While describing the methodology (section 3), there is no connection made between the statistics and the physical flood characteristics? For example, what are the classes (in the classification of section 3.1) deal with?

5) The paper fails to explain the physical relationship between the "Influencing factors" (Table 1) and the flood processes. And why were those particular factors selected? How about antecedent soil moisture and other potential factors?

6) Poor writing throughout. The following is partial list. - L14: to facilitate - L20: cause heavy loss of - L23-24 is that number refers to annual deaths? - L26: the country - L28: 60% of the area in the country is ... a report produced by - L33: It is possible - L47: what does "sceintific manner" means? - L51-52: that is not an accurate description of Dottori etal., because they also provide a water depth. The are based on physical models as well. - L63: can yield - L77: Wha tis that exactly the limitations of the hydrolgical models? And what is the limitations of the proposed method? - L109: by far a heavly affected - L110: located between - L112: Doesn't watershed include mountains and rivers? - L119: have been damaged... must be relocated - L125; reasonable strategy - many more language corrections through out the text! - A more common terms in flood community such as probability of detection and false alarm ratio (rate) can be used - Remove the background color from figure 2 (the region outside of the study region

should be white)

---

## Referee Comment (RC2) · Anonymous Referee #2 · 4 Mar 2017

General comments: Overall the presented work is technically interesting and contains novelties. However, major revisions are required to make it suitable for publication in GMD. The paper currently does not clarify the benefits of the proposed data-intensive model for management purposes. The presented results seem promising in the region of study but general statements about superiority of the proposed model in comparison with other techniques could only be made through evaluation in other flood prone areas. In terms of presentation and English writing, the paper is quite poor in its current form and does not seem suitable for publication without major edition.

[Figure]

Specific comments: 1. Abstract is too short and not informative. 2. The literature review of flood forecasting is poor. Current literature review is only focused on specific studies similar to the current work while ignoring the overall picture of flood and streamflow forecasting. 3. What is the definition of flood used in this study? What is the difference between flood and no-flood? How severe an event needs to be to be called flood? How can predictions be useful for government agencies without providing an estimation of the magnitude and severity of the events? 4. Flood points are used in this study, and not flood areas, with GIS maps. So for information on flood influencing factors in each of these flood points, how many map pixels were used? Was each flood point only associated with the pixel it was located in? If more map pixels than one were used to get information on flood influencing factors for each flood point, how was the area of analysis (relevant pixels) determined for each flood point?

Technical corrections: There are too many instances of poor English writing throughout the paper to be listed. Major edition seems necessary to make the paper suitable for publication.

---

## Author Comment (AC1) · 15 Mar 2017

Authors' reply to Reviewer 1's comments

Journal Name: Geosci. Model Dev Manuscript: Bayesian Framework Based on Gaussian Mixture Model and Radial Basis Function Fisher Discriminant Analysis for Flood Spatial Prediction (BayGmmKda V1.1)

Reply to Reviewer's Comments

Reviewer's comment: The authors use a data-driven tool to establish a relationship

between 10 "flood influencing factors" and the flood itself in a district in Vietnam. While there is a merit in experimenting with statistical tools for trying understand flood occurrence, the current version of the manuscript lacks to demonstrate such a value in using data-driven algorithms for flood prediction. particularly, I have concern with the description and evaluation of the tool. Unless a substantial improvement is made both in the methodology framing and presentation of the manuscript, I would not decremented for publication on GMD.

Response to reviewer's comment: We thank the Reviewer 1 for giving times and expertise to constructively comment on our manuscript. To address your concerns, we have carefully revised and made a substantial improvement in the description and evaluation of the tool. In addition, the methodology framing and presentation of the manuscript have been carefully checked and improved. We believe that the manuscript is a meaningful contribution to the literature because this is the first time the BayGmmKda tool is proposed for flood study with very promising results.

1) There is no clear objective of the work. And it is not clear how the "tool" can be used for flood mapping or prediction (?). A more focused and tailored description of the tool would be helpful to understand and potentially use for the readers of GMD.

Response to reviewer's comment: The objective of the work is to construct a probabilistic model, named as BayGmmKda, for spatial modeling and prediction of flood in Central Vietnam. This region has been critically damaged by floods in recent years due to climate changes and poor land planning. Thus, this model can be very useful since it helps to accurately and reliably construct a flood susceptibility map for this region. Another objective is to employ advances machine learning algorithms including the Gaussian mixture model with the expectation maximization as well as unsupervised training methods and the Radial Basis Function Fisher Discriminant Analysis. The superiority of the proposed model is demonstated via comparisons with previously used machine learning approaches including metaheuritic-trained Adaptive neuro fuzzy inference system, Support Vector Machine, and Bayesian classifier.

In this study, prediction of flood zones relies on an assumption that future flood events are governed by the very similar conditions of flooded zones in the past (Tehrany et al. 2015; Tien Bui et al. 2016). Thus, past records of flood occurrences, coupled with conditioning factors of the areas, are employed as data instances that help to establish the probabilistic model. We formulate the flood assessment problem as a supervised learning task. Therefore, the data samples collected in the past are employed to train the proposed BayGmmKda. With the model structure identified through the training phase, the model can then be used to make assessment on the flood susceptibility for all studied region. The probabilistic model is coded in Matlab enivronment as an easy-to-use toolbox to assist decision makers in flood prediction.

The application of the tool as well as it practical usefulness are demonstrated in the section 5.2 and 5.3 of the manuscript. In these two sections, the model's outstanding accuracy is clearly shown and the flood susceptibility map of the studied region constructed by the tool is demonstrated. Thus, we believe the tool can also be a promising alternative for similar tasks in other studied regions. Based on the reviewer's suggestion, we will address the reviewer's concern by adding more focused and clear decription of the BayGmmKda tool in the revised version of the manuscript for the sake of GMD's readers.

2) The abstract is too short and lacks details of what they attempt to do.

Response to reviewer's comment: Thanks for your comment. We will extend the abstract to describe the study with more details.

3) There is no definition of flood/no flood. In fact it is not clear at all as to what flood is made in the paper. I think flood extent maps should be used for the evaluation, instead of just the selected points. As it is currently used, then streamflow should be used for the evaluation.

Response to reviewer's comment: We thank the reviewer for the comment and would like to explain to you as follows:

[Figure]

Flood points are flood locations that occurred in the study areas, and have been determined based on documentary sources of the Tuong Duong district and interpretation of Landsat 8 Operational Land Imagery. Using DEM, these flood areas were converted to flood points. In addition, flood locations were collected during field works using handhold GPS. A total of 76 flood locations that occurred during the last five years were prepared.

Non-flood points were randomly generated from non-flood areas within the study area based on DEM, i.e. ridges (we has used DEM to generate topographical shades i.e. flat, Ridge, Saddle Ravine, Convex hillside, Saddle hillside, Slope hillside, Concave hillside, Inflection hillside).

Because the above information is available in our previous paper published in Journal of Hydrology, we have provided a citation for this reference in section 3.1 Flood inventory map and flood conditioning factors of the study area) within the revised manuscript. We copy the text here for your review:

"In this study, the flood inventory map established by Tien Bui et al. (2016) was used to analyse the relationships between flood occurrences and influencing factors"

Regarding your comment "it is not clear at all as to what flood is made in the paper", all the floods in this study are flash flood. This is the main flood type in this study area due to characteristics of the terrain. Moreover, we use flood points because flood extent maps are not available. Thus, we employs flood points provided from the sources of local authority and handhold GPS.

Regarding your comment on the streamflow being used for the flood evaluation, in fact, we have performed a literature review on the use of streamflow for evaluating flood susceptibility. However, we found no relevant or feasible guidances to construct the flood susceptibility model for the studied area based on the available data. Thus, we'd like to consider the possibility of using streamflow for flood evaluations in a future research. This direction will be stated in the conclusion of the revised version.

4) While describing the methodology (section 3), there is no connection made between the statistics and the physical flood characteristics? For example, what are the classes (in the classification of section 3.1) deal with?.

Response to reviewer's comment: We'd to thank reviewer for these comments and we totally agree with the reviewer's opinion at this point. We will provide explanations on the connection made between the statistics and the physical flood characteristics in the beginning of the section 3 of the revised manuscript. We copy the texts in the revised manuscript for your review: "The flood modeling in this study is considered to be a binary classification problem within which 'flood' and 'non-flood' are the two class labels of interest. As a result, the probability of pixels belonging to the flood class, which are derived from the model, will be used as susceptibility indices. These susceptibility indices of the pixels are then used to generate the flood susceptibility map."

5) The paper fails to explain the physical relationship between the "Influencing factors" (Table 1) and the flood processes. And why were those particular factors selected?. How about antecedent soil moisture and other potential factors?

Response to reviewer's comment: We agree with the reviewer on this comment. Based on the reviewer's comment, we have provided texts in the revised manuscript with the pertinent reference to explain the physical relationship between the "Influencing factors" (Table 1) and the flood processes as well as the reason why we choose those particular influencing factors. We copy the texts from the revised manuscript here for your review: "In our previous works of Tien Bui et al. (2016), the physical relationships between influencing factors and flood processes were analyzed. Based on the findings, a total of ten influencing factors were selected in this study, including slope (o), elevation(m), curvature, TWI, SPI, distance to river (m), stream density (km/km2), NDVI, lithology, and rainfall (mm)."

Regarding the comment "How about antecedent soil moisture and other potential factors?", we'd like to explain as follows: In fact, the selection of the conditioning factors

varies from one study area to another based on different characteristics of each place. One variable can have high degree of impact in flooding in a specific area, but it can be without any influence in another regions (Kia et al. 2012). In this study, due to the data availability, we have not employed antecedent soil moisture as a conditioning variable. However, we appreciate the reviewer's suggestions and we think that further studies should be carried out to investigate the influences of antecedent soil moisture and other potential factors for the study regions in Vietnam. This point will be addressed in our conclusion in the revised version.

6) Poor writing throughout. The following is partial list. - L14: to facilitate - L20: cause heavy loss of - L23-24 is that number refers to annual deaths? - L26: the country - L28: 60% of the area in the country is ... a report produced by - L33: It is possible - L47: what does "sceintific manner" means? - L51-52: that is not an accurate description of Dottori etal., because they also provide a water depth. The are based on physical models as well. - L63: can yield - L77: Wha tis that exactly the limitations of the hydrolgical models? And what is the limitations of the proposed method? - L109: by far a heavly affected - L110: located between - L112: Doesn't watershed include mountains and rivers? - L119: have been damaged... must be relocated - L125; reasonable strategy - many more language corrections through out the text! - A more common terms in flood community such as probability of detection and false alarm ratio (rate) can be used - Remove the background color from figure 2 (the region outside of the study region should be white)

Response to reviewer's comment: We'd like to thank the reviewer for your great help. All addressed grammatical and presentation issues will be addressed in the revised version. The whole manuscript has been proofread to improve the writing.

Reference Kia, M. B., Pirasteh, S., Pradhan, B., Mahmud, A. R., Sulaiman, W. N. A., and Moradi, A. (2012). "An artificial neural network model for flood simulation using GIS: Johor River Basin, Malaysia." Environ Earth Sci, 67(1), 251-264. Tehrany, M. S., Pradhan, B., Mansor, S., and Ahmad, N. (2015). "Flood susceptibility assessment

using GIS-based support vector machine model with different kernel types." CATENA, 125, 91-101. Tien Bui, D., Pradhan, B., Nampak, H., Bui, Q.-T., Tran, Q.-A., and Nguyen, Q.-P. (2016). "Hybrid artificial intelligence approach based on neural fuzzy inference model and metaheuristic optimization for flood susceptibilitgy modeling in a high-frequency tropical cyclone area using GIS." J. Hydrol., 540, 317-330.
* * *

---

## Author Response (AR1)

**Authors' Response**

**Journal Name: Geosci. Model Dev**

Manuscript: **Bayesian Framework Based on Gaussian Mixture Model and Radial Basis Function Fisher Discriminant Analysis for Flood Spatial Prediction (BayGmmKda V1.1)**

The authors would like to thank the reviewers for the useful and constructive comments and suggestions on our manuscript. We have strictly reviewed and revised the manuscript accordingly, and detailed corrections are listed below point by point. The modifications of the manuscript according to the comments of the reviewers are highlighted in the tracked change version of the revised manuscript.

The manuscript has been resubmitted to your journal. We look forward to your positive response.

Sincerely,

**Authors' reply to Reviewer 1's comments**

**Reply to Reviewer's Comments**

**Reviewer's comment**: *The authors use a data-driven tool to establish a relationship between 10 "flood influencing factors" and the flood itself in a district in Vietnam. While there is a merit in experimenting with statistical tools for trying understand flood occurrence, the current version of the manuscript lacks to demonstrate such a value in using data-driven algorithms for flood prediction. particularly, I have concern with the description and evaluation of the tool. Unless a substantial improvement is made both in the methodology framing and presentation of the manuscript, I would not decremented for publication on GMD.*

**Response to reviewer's comment**: We thank the **Reviewer 1** for giving times and expertise to constructively comment on our manuscript. To address your concerns, we have carefully revised and made a substantial improvement in the description and evaluation of the tool. In addition, the methodology framing and presentation of the manuscript have been carefully checked and improved. We believe that the manuscript is a meaningful contribution to the literature because this is the first time the BayGmmKda tool is proposed for flood study with very promising results.

*1) There is no clear objective of the work. And it is not clear how the "tool" can be used for flood mapping or prediction (?). A more focused and tailored decription of the tool would be helpful to understand and potentially use for the readers of GMD.*

**Response to reviewer's comment**:

The objective of the work is to construct a probabilistic model, named as BayGmmKda, for spatial modeling and prediction of flood in Central Vietnam. This region has been critically damaged by floods in recent years due to climate changes and poor land planning. Thus, this model can be very useful since it helps to accurately and reliably construct a flood susceptibility map for this region. Another objective is to employ advances machine learning algorithms including the Gaussian mixture model with the expectation maximization as well as unsupervised training methods and the Radial Basis Function Fisher Discriminant Analysis. The superiority of the proposed model is demonstated via comparisons with previously used machine learning approaches including metaheuritic-trained Adaptive neuro fuzzy inference system, Support Vector Machine, and Bayesian classifier.

In this study, prediction of flood zones relies on an assumption that future flood events are governed by the very similar conditions of flooded zones in the past (Tien Bui et al., 2016;Tehrany et al., 2015). Thus, past records of flood occurrences, coupled with conditioning factors of the areas, are employed as data instances that help to establish the probabilistic model. We formulate the flood assessment problem as a supervised learning task. Therefore, the data samples collected in the past are employed to train the proposed BayGmmKda. With the model structure identified through the training phase, the model can then be used to make assessment on the flood susceptibility for all studied region. The probabilistic model is coded in Matlab enivronment as an easy-to-use toolbox to assist decision makers in flood prediction. The application of the tool as well as it practical usefulness are demonstrated in the section 5.2 and 5.3 of the manuscript. In these two sections, the model's outstanding accuracy is clearly shown and the flood susceptibility map of the studied region constructed by the tool is demonstrated.

Thus, we believe the tool can also be a promising alternative for similar tasks in other studied regions.

Based on the reviewer's suggestion, we have addressed the reviewer's concern by adding more focused and clear decription of the BayGmmKda tool in the revised version of the manuscript for the sake of GMD's readers. We copy the text from the revised manuscript (page 3) for your reference:

"To construct flood susceptibility evaluation models, databases of GIS that contains a set of flood influencing factors and information of past flood events is established at the first step. At the next step, advanced soft computing models can be utilized to distinguish the flood vulnerable areas for the entire studied region (Tehrany et al., 2015b). In this way, the flood prediction problem boils down to a supervised classification task. Nevertheless, most models in the current studies can only produce qualitative outputs of flood prediction outcome (i.e. flood–no flood) (Tien Bui et al., 2016b;Tehrany et al., 2015b); probabilistic evaluations have rarely been seen in the literature. Given these motivations, the objective of this study is to construct a probabilistic model for spatial prediction of flood in Central Vietnam. The newly proposed method aims at enhancing the prediction accuracy as well as deriving probabilistic evaluations of flood susceptibility in a regional scale. The derived flood susceptibility is of great usefulness for local authorities in land-use planning and management. The local authorities may overlay the flood susceptibility map onto planed land-use maps in different scenarios."

*2) The abstract is too short and lacks details of what they attempt to do.*

**Response to reviewer's comment**: Thanks for your comment. We have extend the abstract to describe the study with more details as follows:

"**Abstract.** In this study, a probabilistic model, named as BayGmmKda, is proposed for flood assessment with a study area in Central Vietnam. The new model is a Bayesian framework constructed by a combination of Gaussian Mixture Model (GMM), Radial Basis Function Fisher Discriminant Analysis (RBFDA), and a Geographic Information System database. To compute the posterior probability of flood, the GMM algorithm is utilized for modeling the data distributionsof flood conditioning factors. Additionally, the RBFDA method is employed in BayGmmKda to construct a latent variable that maximizes the data discrimination with respect to the two class labels of 'flood' and 'no-flood'. Experiments used for measuring the model performance point out that the proposed hybrid framework is superior to other benchmark models including the adaptive neuro fuzzy inference system and the support vector machine. To facilitate the model implementation, a software program of BayGmmKda has been developed in Matlab. The BayGmmKda program can accurately establish a flood susceptibility map for the study region. Accordingly, local authorities can overlay this susceptibility map onto various land-use maps for the purpose of land-use planning or management. "

*3) There is no definition of flood/no flood. In fact it is not clear at all as to what flood is made in the paper. I think flood extent maps should be used for the evaluation, instead of just the selected points. As it is currently used, then streamflow should be used for the evaluation.*

**Response to reviewer's comment**: We thank the reviewer for the comment and would like to explain to you as follows:

Flood points are flood locations that occurred in the study areas, and have been determined based on documentary sources of the Tuong Duong district and interpretation of Landsat 8 Operational Land Imagery. Using DEM, these flood areas were converted to flood points. In addition, flood locations were collected during field works using handhold GPS. A total of 76 flood locations that occurred during the last five years were prepared.

Non-flood points were randomly generated from non-flood areas within the study area based on DEM, i.e. ridges (we has used DEM to generate topographical shades i.e. flat, Ridge, Saddle Ravine, Convex hillside, Saddle hillside, Slope hillside, Concave hillside, Inflection hillside).

Because the above information is available in our previous paper published in Journal of Hydrology, we have provided a citation for this reference in section 3.1 Flood inventory map and flood conditioning factors of the study area) within the revised manuscript. We copy the text here for your review:

"In this study, the flood inventory map established by Tien Bui et al. (2016) was used to analyse the relationships between flood occurrences and influencing factors"

Regarding your comment "*it is not clear at all as to what flood is made in the paper*", all the floods in this study are flash flood. This is the main flood type in this study area due to characteristics of the terrain. This information has been provided on page 6 of the revised manuscript:

"The flood inventory map stores documentations of past flood events (see **Figure 1**). It is noted that the type of floods in this study area is flash flood. This is the main flood type in this region due to characteristics of the terrain."

Moreover, we use flood points because flood extent maps are not available. Thus, we employs flood points provided from the sources of local authority and handhold GPS.

Regarding your comment on the streamflow being used for the flood evaluation, in fact, we have performed a literature review on the use of streamflow for evaluating flood susceptibility. However, we found no relevant or feasible guidances to construct the flood susceptibility model for the studied area based on the available data. Thus, we'd like to consider the possibility of using streamflow for flood evaluations in a future research. This direction has been stated in the conclusion of the revised version.

*4) While describing the methodology (section 3), there is no connection made between the statistics and the physical flood characteristics? For example, what are the classes (in the classification of section 3.1) deal with?.*

**Response to reviewer's comment**:

We'd to thank reviewer for these comments and we totally agree with the reviewer's opinion at this point. We have provided explanations on the connection made between the statistics and the physical flood characteristics in the beginning of the section 3 of the revised manuscript. We copy the texts in the revised manuscript for your review:

"The flood modeling in this study is considered to be a binary classification problem within which 'flood' and 'non-flood' are the two class labels of interest. As a result, the probability of pixels belonging to the flood class, which are derived from the model, will be used as susceptibility indices. These susceptibility indices of the pixels are then used to generate the flood susceptibility map."

5) The paper fails to explain the physical relationship between the "Influencing factors" (Table 1) and the flood processes. And why were those particular factors selected?. How about antecedent soil moisture and other potential factors?

**Response to reviewer's comment**:

We agree with the reviewer on this comment. Based on the reviewer's comment, we have provided texts in the revised manuscript with the pertinent reference to explain the physical relationship between the "Influencing factors" (Table 1) and the flood processes as well as the reason why we choose those particular influencing factors. We copy the texts from the revised manuscript on page 7 here for your review:

"Based on the previous work of Tien Bui et al. (2016b), the physical relationships between influencing factors and flood processes have been analyzed. Accordingly, a total of ten influencing factors were selected in this study; they include slope ($IF_1$), elevation ($IF_2$), curvature ($IF_3$), topographic wetness index (TWI) ($IF_4$), stream power index (SPI) ($IF_5$), distance to river ($IF_6$), stream density ($IF_7$), normalized difference vegetation index (NDVI) ($IF_8$), lithology ($IF_9$), and rainfall ($IF_{10}$). These factors are used to analyze the flood vulnerability for the studied area and a GIS database consisting of the flood inventory map and the chosen factors has been established."

Regarding the comment "How about antecedent soil moisture and other potential factors?", we'd like to explain as follows:

In fact, the selection of the conditioning factors varies from one study area to another based on different characteristics of each place. One variable can have high degree of impact in

flooding in a specific area, but it can be without any influence in another regions (Kia et al., 2012). In this study, due to the data availability, we have not employed antecedent soil moisture as a conditioning variable. However, we appreciate the reviewer's suggestions and we think that further studies should be carried out to investigate the influnences of antecedent soil moisture and other potential factors for the study regions in Vietnam. This point has been addressed in our conclusion in the revised version as follows:

"Result comparison pointed out that the BayGmmKda is superior to other benchmark approaches including ANFIS and SVM. Therefore, the proposed model, featured by its high predictive accuracy and the capability of delivering probabilistic outputs, is a promising alternative for flood susceptibility prediction. Neverthless, the proposed method also suffers from seraval drawbacks. BayGmmKda is a data-driven tool; thus, field works at the studied area and data analyses from remote sensing are necessary for the model construction phase. These data collecting and analyzing can be time-consuming. Furthermore, a grid search procedure is used for hyper-parameter setting and this process also requires a high computational cost especially for large-scale data sets. In addition, the outcome of this grid search procedure may not be optimal; more advanced mode selection approaches (e.g. metaheuristics) can be utilized to further improve the model accuracy. Future extensions of this research may include the model application in flood prediction for other study areas, investigations of other flood conditioning factors (e.g. streamflow and antecedent soil moisture) which may be relevant for flood analysis, and improving the current model with other novel soft computing methods (e.g. feature selection, pattern classification, dimension reduction, metaheuristic optimization, etc.). to alleviate the aforementioned drawbacks as well as to enhance the model performance."

*6) Poor writing throughout.*

*The following is partial list. - L14: to facilitate - L20: cause heavy loss of - L23-24 is that number refers to annual deaths? - L26: the country - L28: 60% of the area in the country is ... a report produced by - L33: It is possible - L47: what does "sceintific manner" means? - L51-52: that is not an accurate description of Dottori etal., because they also provide a water depth. The are based on physical models as well. - L63: can yield - L77: Wha tis that exactly the limitations of the hydrolgical models? And what is the limitations of the proposed method? - L109: by far a heavly affected - L110: located between - L112: Doesn't watershed include mountains and rivers? - L119: have been damaged... must be relocated - L125; reasonable strategy - many more language corrections through out the text! - A more common terms in flood community such as probability of detection and false alarm ratio (rate) can be used - Remove the background color from figure 2 (the region outside of the study region should be white)*

**Response to reviewer's comment**:  We'd like to thank the reviewer for your great help. All addressed grammatical and presentation issues have been addressed in the revised version. The whole manuscript has been proofread to improve the writing. Please refer to the tracked change version for the highlighted corrections.

Based on your comments, we have provided the limitation of hydrological models on page 4 as follows:

    "Because of the criticality of flood prediction, this problem has gained an increasing attention from the academic community. Following this trend, various flood analyzing tools have been developed, ranging from relatively simple methods to more sophisticated methodologies involving hydrological and hydraulic models (Winsemius et al., 2013;Papaioannou et al., 2015). In general, the goal of constructing hydrological models is to acquire an accurate evaluation of

discharge over the watersheds. Moreover, it is noted that to establish such models, large-scale field works and deployments of measuring equipments are necessary for collecting data (Fenicia et al., 2008). A review done by Sanyal and Lu (2004) pointed out that the density of gauging stations in developing countries is very low and this fact imposes a great obstacle for establishing accurate hydrological models. In addition, the complex and nonlinear nature of the flood modeling problem also bring about difficulties for hydrological methods and techniques (Sahoo et al., 2006)."

The limitations of the proposed model have been stated as follows in the conclusion:

"Result comparison pointed out that the BayGmmKda is superior to other benchmark approaches including ANFIS and SVM. Therefore, the proposed model, featured by its high predictive accuracy and the capability of delivering probabilistic outputs, is a promising alternative for flood susceptibility prediction. Neverthless, the proposed method also suffers from seraval drawbacks. BayGmmKda is a data-driven tool; thus, field works at the studied area and data analyses from remote sensing are necessary for the model construction phase. These data collecting and analyzing can be time-consuming. Furthermore, a grid search procedure is used for hyper-parameter setting and this process also requires a high computational cost especially for large-scale data sets. In addition, the outcome of this grid search procedure may not be optimal; more advanced mode selection approaches (e.g. metaheuristics) can be utilized to further improve the model accuracy."

**Authors' reply to Reviewer 2's comments**

**Reply to Reviewer's Comments**

**General comment**

*Overall the presented work is technically interesting and contains novelties. However, major revisions are required to make it suitable for publication in GMD.*

**Response to reviewer's comment:**

We thank the reviewer for taking times and expertise to constructively comment on our manuscript. We believe that the manuscript is a meaning full contribution to the body of knowledge because this is the first time the BayGmmKda model is proposed for flood study with very promising result. We have carefully revised and provided a substantial improvement for the revised manuscript according to the reviewer's suggestions.

Regarding your comment "*The paper currently does not clarify the benefits of the proposed data-intensive model for management purposes.*", we 'd like to reply as follows:

The model proposed in this study can make inference on the spatial prediction of flood or flood susceptibility. The current model cannot provide estimation of the magnitude and severity the events. Nevertheless, the flood susceptibility assessment is of great usefulness for local authorities in landuse planning and management by overlaying the flood susceptibility map onto planed land use maps in different scenarios. Accordingly, areas with very high flood susceptibility could be determined. Of cause, the current flood susceptibility map is only in district scale; therefore, after determining high flood susceptibility areas, larger scale studies should be carried out focusing on these areas. Based on the reviewer's comment, we have better

addressed the benefits of the proposed model for land-use management purposes in the introduction part of the revise version as follows on page 3:

" Given these motivations, the objective of this study is to construct a probabilistic model for spatial prediction of flood in Central Vietnam. The newly proposed method aims at enhancing the prediction accuracy as well as deriving probabilistic evaluations of flood susceptibility in a regional scale. The derived flood susceptibility is of great usefulness for local authorities in land-use planning and management. The  local authorities may overlay the flood susceptibility map onto planed land-use maps in different scenarios."

Regarding your comment "*The presented results seem promising in the region of study but general statements about superiority of the proposed model in comparison with other techniques could only be made through evaluation in other flood prone areas.*", we 'd like to reply as follows:

We agree with the reviewer on this comment. The model proposed in this study will be applied for spatial modeling of flood in other study areas in Vietnam as well as in other countries. However, the data collection and processing are time-consuming. Thus, we consider this comment of the reviewer as a future research direction. We have modified the conclusion in the revised version to address this comment of the reviewer.

Regarding the comment  "In terms of presentation and English writing, the paper is quite poor in its current form and does not seem suitable for publication without major edition.", We have carefully checked and improved the English writing of the revised manuscript.

**Specific comments**

1. Abstract is too short and not informative.

**Response to reviewer's comment:** We agree with the reviewer, therefore we have rewritten the abstract in the revised manuscript as follows:

"**Abstract.** In this study, a probabilistic model, named as BayGmmKda, is proposed for flood assessment with a study area in Central Vietnam. The new model is a Bayesian framework constructed by a combination of Gaussian Mixture Model (GMM), Radial Basis Function Fisher Discriminant Analysis (RBFDA), and a Geographic Information System database. To compute the posterior probability of flood, the GMM algorithm is utilized for modeling the data distributionsof flood conditioning factors. Additionally, the RBFDA method is employed in BayGmmKda to construct a latent variable that maximizes the data discrimination with respect to the two class labels of 'flood' and 'no-flood'. Experiments used for measuring the model performance point out that the proposed hybrid framework is superior to other benchmark models including the adaptive neuro fuzzy inference system and the support vector machine. To facilitate the model implementation, a software program of BayGmmKda has been developed in Matlab. The BayGmmKda program can accurately establish a flood susceptibility map for the study region. Accordingly, local authorities can overlay this susceptibility map onto various land-use maps for the purpose of land-use planning or management. "

2. The literature review of flood forecasting is poor. Current literature review is only focused on specific studies similar to the current work while ignoring the overall picture of flood and streamflow forecasting.

**Response to reviewer's comment:** We understand that the overall picture of flood and streamflow forecasting is very large for the objective of this particular study. After performing literature review, we see that there are two groups of approaches for flood and streamflow forecasting: (i) the first one is "regression modeling" and (ii) the second one is "classification modeling"

The first approach group has used for very long time, but required detailed monitoring data for modeling, these data are difficult to obtain for Vietnam as a developing country. The modeling result of this approach group could provide spatial and temporal prediction of flood for study areas.

The second group is relatively new and does not require flood monitoring data. It uses "on – flood pixel" and "off - non flood pixel" for flood modeling. Therefore, this approach is feasible for modeling of large areas with the use of remote sensing and GIS data. In other words, the input-output datasets in this approach is very different with those of the traditional approaches. The modeling result of this approach group provide only where flood may occur (spatial prediction of flood or flood susceptibility), this does not provide temporal prediction or flood discharge.

In this study, we use the second approach group, therefore we have mainly specific studies similar to our works and we think that the current literature review is reasonable.

3. What is the definition of flood used in this study? What is the difference between flood and no-flood? How severe an event needs to be to be called flood? How can predictions be useful for government agencies without providing an estimation of the magnitude and severity of the events?

**Response to reviewer's comment:**

Floods in this study are flood locations that occurred in the study areas and have been determined based on documentary sources of the local district, interpretation of Landsat 8 Operational Land Imagery. In addition, flood locations were collected during field works using handhold GPS. Non-flood points were randomly generated from non-flood areas within the study area based on DEM, i.e. ridges (we has used DEM to generate topographical shades i.e. flat, Ridge, Saddle Ravine, Convex hillside, Saddle hillside, Slope hillside, Concave hillside, Inflection hillside).

Regarding your comment "How severe an event needs to be to be called flood?", we 'd like to reply as follows: In this study, only flash flood is modelled. The flood locations used in this study were provided by the local authority. With the current database, the exact information on the severity of these floods is not available. However, all 76 floods in this study caused huge damages to the local people.

The information regarding the determination of flood point in the studied region has been provided more clearly on page 7:

"The flood inventory map stores documentations of past flood events (see **Figure 1**). It is noted that the type of floods in this study area is flash flood. This is the main flood type in this region due to characteristics of the terrain.The map was constructed by gathering information of the study area, field works at flood areas, and analyses from results of the Landsat 8 Operational Land Imagery (from 2010 to 2014) with the resolution of 30m (retrieved from http://earthexplorer.usgs.gov). Furthermore, the location of flood events was also verified by field works carried out in 2014 with handhold GPS devices. In summary, the total number of

flood locations during the last five years was recorded to be 76. It is noted that flood locations were determined by overlaying the flood polygons in the inventory map and the Digital Elevation Model (DEM). Moreover, only pixels in the map that associate with flood points are used to extract the influencing factors used for flood prediction."

Regarding the comment, "How can predictions be useful for government agencies without providing an estimation of the magnitude and severity of the events? ",we'd like to response as follows: As we explained in the above answer. The flood model in this study could provide only where flood may occur called spatial prediction of flood or flood susceptibility. The current model is not able to deliver estimation of the magnitude and severity the events. The flood susceptibility is still useful for local authorities in landuse planning and management by overlaying the flood susceptibility map to planed land use maps in different scenario. Subsequently, areas with very high flood susceptibility could be determined. Of course, the flood susceptibility map is only in district scale; therefore, after determining high flood susceptibility areas, larger scale studies should be carried out focusing on these areas.

4. Flood points are used in this study, and not flood areas, with GIS maps. So for information on flood influencing factors in each of these flood points, how many map pixels were used? Was each flood point only associated with the pixel it was located in? If more map pixels than one were used to get information on flood influencing factors for each flood point, how was the area of analysis (relevant pixels) determined for each flood point?

**Response to reviewer's comment:**

We have 76 flood polygons in the flood inventory map. However, we used 76 points for these polygons and these points were determined based on overlaying these polygons and the DEM. The grid (30 m size) of the district map was constructed by 3900 columns and 4125 rows.

Regarding the question "Was each flood point only associated with the pixel it was located in?. If more map pixels than one were used to get information on flood influencing factors for each flood point, how was the area of analysis (relevant pixels) determined for each flood point?": Yes, each flood point only associated with the pixel it was located in. No more map pixels than one were used to get information on flood influencing factors for each flood point. More details on the determination of flood points are provided on page 7as follows:

"It is noted that flood locations were determined by overlaying the flood polygons in the inventory map and the Digital Elevation Model (DEM). Moreover, only pixels in the map that associate with flood points are used to extract the influencing factors used for flood prediction."

**Technical corrections**

There are too many instances of poor English writing throughout the paper to be listed.

Major edition seems necessary to make the paper suitable for publication.

**Response to reviewer's comment:** According to the reviewer's comment, the whole manuscript has been be proofread to improve the English writing.

**Reference**

[revised manuscript text omitted]

---

## Author Response (AR2)

**Comments from Editor**

**Comment 1**: Reviewer #1 is concerned with the description of the methodology and notes that this comment was the same in their first review. Please consider how the presentation of the method might be improved for the reader. Perhaps a flow diagram like Figure 5 in section 3.2 that sets out the components of the method prior to the technical discussion would help the reader?

**Response**

We agree with the Editor's comments and have provided a new Figure 3 in section 3.2 according to the suggestion of the Editor. Furthermore, we have revised the Figure 6 for better describing the components of the proposed model.

[Figure]

**Figure 3** General concept of the Bayesian Framework for flood classification

[Figure]

**Figure 6** The proposed BayGmmKda

The section 4.2 on page 17 has been revised to describe the model concept more clearly as follows:

"Firstly, the whole dataset, including 152 data samples, was separated into two sets: Training Set (90% or 137 samples) employed for model establishing and Testing Set (10% or 15 samples) used for model testing. It is noted that the input variables of the dataset have been normalized using the Min-Max normalization; the purpose of data normalization was to hedge against the situation of unbalanced variable magnitudes.

Secondly, a latent input factor was generated using the RBFDA (explained in section 3.4) and added to the training dataset aiming to enhance the classification performance. Subsequently, the feature evaluation was performed to quantify the degree of relevance of each input factors with the flood inventories in the Training Set. Any non-relevant factor should be eliminated from the modeling process to reduce noise and enhance the model performance (Tien Bui et al., 2016a;Tien Bui et al., 2017). For this purpose, in this research, the Mutual Information Criterion (Kwak and Choi, 2002;Hoang et al., 2016), a widely employed techniques for feature selection in machine learning, was selected to express the pertinence of each influencing factor to the flood. It is noticed that the larger the mutual information, the stronger the relevancy between the influencing factor and flood.

In the next step, the BayGmmKda model was trained and established using the Training Set. The purpose of the training process was to find the best parameters for the mixture component ($k$) used in GMM and the kernel function bandwidth ($\dagger$) used in RBFDA of the BayGmmKda model. To determine the best $k$, the EM algorithm that employs Akaike Information Criterion (AIC) (Akaike, 1974) was used. Thus, the value of $k$ was varied from 1 to 20, and then, AIC was estimated and used to select the model that exhibits the best fit to the data at hand. It is noted that a model with a few number of mixture components ($k$) indicates a less degree of complexity (Olivier et al., 1999). In addition, the unsupervised GMM learning (Figueiredo and Jain, 2002) is also used for autonomously determining the best $k$. Accordingly, the model starts with a maximum component number ($k$) of 20; the algorithm carries out the model selection process by removing irrelevant mixture components if applicable. To determine the best $\dagger$, the Grid Search procedure is performed and the parameter $\dagger$ corresponding to the highest classification accuracy rate was selected.

Using the best $k$ and $\dagger$ in the previous step, the final BayGmmKda model was finally constructed and the Bayesian classification framework was derived. The Bayesian framework was then used to estimate the posterior probability (flood susceptibility index) for all the pixels in the study areas. The flood susceptibility index was then transferred to a raster format to open in ArcGIS."

**Comment 2:** Finally, I agree with reviewer #1 comment regrading hydrological modelling and I wonder if you could also explicitly comment on 1) the record length required by your technique and 2) the potential for data gaps to be a problem, either between Landsat overpasses or due to cloud cover. Perhaps this is intrinsic to the test you have run however a comment on these factors, that are often critical for optical remote sensing based methods, would be a valuable addition to the conclusions I believe.

**Response**

We agree with the editor and the reviewer. Therefore, we have rewritten the texts in the revised manuscript. We copy these texts from page 2 here for review:

"To predict flood occurrence, conventional approaches require time series of meteorological and streamflow data at gauging stations (Machado et al., 2015). However, this is difficult for many areas in developing countries where no gauging station is available. Therefore, new modeling approaches should be explored and investigated. Given these motivations, this study proposes a novel methodology designed for enhancing the prediction accuracy as well as deriving probabilistic evaluation of flood susceptibility in a regional scale. Accordingly, spatial prediction of flood is carried out based on a statistical assumption that flood in the future will occur under the same conditions that triggered them in the past (Tien Bui et al., 2016b). In this way, the flood prediction problem boils down to an on-off supervised classification task, where the flood inventories are used as a flood class, whereas a non-flood class is derived from areas that have not yet damaged by flood. Consequently, spatial prediction of flood is derived based on probability of study area pixels belonging to the flood class. To yield probabilistic outputs of flood, this study proposes, for the first time, a Bayesian framework established based on a Gaussian mixture model (GMM) and the Kernel Fisher Discriminant Analysis (KFDA). GMM is employed for density approximation to calculate the posterior probability of flood (flood susceptibility index), whereas KFDA is employed to construct a latent variable for from the geo-environmental conditions to enhance the performance of the Bayesian model."

- "1) the record length required by your technique"

Response: No record length (time series data) required by our technique because we used new modeling approach that very different with conventional approaches in flood modeling i.e. rainfall-runoff modeling and time series regression. To avoid any misunderstanding in our proposed method, we have provided more texts in section 2 literature review. We copy these texts here for review:

"Because of the criticality of flood prediction, this problem has gained an increasing attention from the academic community. Following this trend, various flood analyzing tools have been developed, ranging from relatively simple methods to more sophisticated methodologies (Winsemius et al., 2013;Papaioannou et al., 2015;Gao et al., 2017). Basically, these tools could be classified into statistical analysis, rainfall-runoff models, and classification models. Statistical analysis uses long-term recorded time series data at gauged stations to establish regression models, and then, the models are used to transform flood information to ungauged basins (Yue et al., 1999;Cunnane, 1988;McCuen, 2016). Thus, these models are capable to provide flood predictions both in space and time. However, long-term data are not always available, and in many cases, they are general too short for reliable estimating of extreme quantiles (Seckin et al., 2013b;Nguyen et al., 2014).

Rainfall–runoff models, which deal with estimating of runoff from rainfall, are considered to be the most extensively used for flood prediction and management (Nayak et al., 2013;Ciabatta et al., 2016;Bennett et al., 2016). Various types of rainfall–runoff models can be found in literature, varying from empirical models to highly sophisticated physical processes. Empirical models could be established based on statistical techniques (Brocca et al., 2011) or advanced machine learning algorithms (Lohani et al., 2011) to model rainfall and runoff using historical time series data. In addition, physical processes models focus on simulating hydrological processes in a basin based on a set of mathematical equations governing physical processes of water flow and surfaces (Chiew et al., 1993;Birkel et al., 2010;Arnold et al., 1998;Beven et al., 1984;Grimaldi et al., 2013). In general, rainfall–runoff models require relative long term time series data at gauging stations. However, the density of gauging stations in developing countries is very low and this fact imposes a great obstacle

for establishing accurate hydrological models (Fenicia et al., 2008). In addition, large-scale field works and deployments of measuring equipment are necessary for collecting data. Nevertheless, the complex and nonlinear nature of the flood modeling problem also bring about difficulties for hydrological methods and techniques (Sahoo et al., 2006).

In recent year, a new flood modeling approach called "on-off" classification of flood occurrence has been successfully proposed for spatial prediction of flood, also called flood susceptibility (Tien Bui et al., 2016d;Tehrany et al., 2014;Tehrany et al., 2015b). Accordingly, no time series data is required for the model calibration and the establishment of flood models is based on flood inventories (flood class) and non-flood areas (non-flood class). Accordingly, the probability of a pixel in the study area belongs to the flood class is used as flood susceptibility index. Although flood susceptibility map provides no temporal prediction or return period of flood, the flood map is capable delineating highly susceptible areas. Thus, it is a powerful flood analysis tool for decision-makers that could be used in landuse planning and flood management. Literature review shows that data-driven methods integrated with GIS databases have demonstrated their effectiveness and accuracy in large scaled flood susceptible predictions. An fuzzy logic based algorithm has been used to develop a map of flooded areas from synthetic aperture radar imagery, used for the operational flood management system in Italia, was established by Pulvirenti et al. (2011). A model based on the frequency ratio approach and GIS for spatial prediction of flooded regions was first introduced by Lee et al. (2012); the spatial database were constructed by field surveys and maps of the topography, geology, land cover, and infrastructure."

- "and 2) the potential for data gaps to be a problem, either between Landsat overpasses or due to cloud cover"

Response: There is no potential problem for data gaps in this research because we do not use time series data as in traditional approaches for flood modeling. We used a new modeling approach and we called this as "*called "on-off" classification approach*", "On" = flood areas and "off" = not-yet flood areas. So study areas were divided in pixels and probability of a pixel belongs to the "flood class" will be used as flood susceptibility index. It means that "Landsat overpasses or due to cloud cover" is not a problem because we do not use Landsat image with cloud cover. Landsat imagery + documents and reports of the local government + DEM help us to find flash food inventories in the study areas. Although Landsat overpasses, but the Landsat imagery still show "flash flood signals" in the images, so extensive fieldworks with handhold GPS were used to collect flood locations. We hope these explanations are very clear to you.

**Comments from Anonymous Referee #1**

**Comment 1:** The manuscript is improved from the previous version. As I noted in the previous review, there is a merit in experimenting with statistical tools for trying understand flood occurrence. The proposed tool is relevant in that sense, and certainly deals with a most important topic of flood risk assessment. I thank the authors for addressing some of the issues raised in the previous review.

**Response**

We thank the Reviewer1 for taking your time and expertise to review our manuscript. We believe that the manuscript is a meaning full contribution to the literature because this is the first time the BayGmmKda tool is proposed for flood study with very promising result.

**Comment 2:** However, there are still several issues that need to be addressed before accepting for publication. The quality of writing hasn't substantially improved, and the description of the methodology (the core part of manuscript really) hasn't changed. I would like to ask the authors to carefully proofread (or ask some else with a better grammar to proofread) the manuscript.

**Response**

We have carefully proofread the manuscript as suggested by the reviewer. We have rewritten the description of the methodology, provided new Figure 3 (Section 3.2 on page 10) and Figure 6 with rewritten explanation (Section 4 on page 16). Please refer to the revised manuscript for more details.

**Comment 3: "**It is true that hydrological modelling has its challenges, but I disagree with the argument presented against hydrological models (under section 2). The limited availability of data limits the utility of models, but the data-driven models (such as the one presented) depend on data more than physics based models. Therefore I disagree that "The new approach based on GIS successfully evades the limitations of the hydrological models" Line 91."

**Response**

After carefully the section 2, we agree with the Reviewer. Therefore, we have provided more texts in literature review and also rewritten texts to avoid misunderstanding. We copy the text from the revised manuscript on page 3 here for your review:

"Because of the criticality of flood prediction, this problem has gained an increasing attention from the academic community. Following this trend, various flood analyzing tools have been developed, ranging from relatively simple methods to more sophisticated methodologies (Winsemius et al., 2013;Papaioannou et al., 2015;Gao et al., 2017). Basically, these tools could be classified into statistical analysis, rainfall-runoff models, and classification models. Statistical analysis uses long-term recorded time series data at gauged stations to establish regression models, and then, the models are used to transform flood information to ungauged basins (Yue et al., 1999;Cunnane, 1988;McCuen, 2016). Thus, these models are capable to provide flood predictions both in space and time. However, long-term data are not always available, and in many cases, they are general too short for reliable estimating of extreme quantiles (Seckin et al., 2013b;Nguyen et al., 2014).

Rainfall–runoff models, which deal with estimating of runoff from rainfall, are considered to be the most extensively used for flood prediction and management (Nayak et al., 2013;Ciabatta et al., 2016;Bennett et al., 2016). Various types of rainfall–runoff models can be found in literature, varying from empirical models to highly sophisticated physical processes. Empirical models could be established based on statistical techniques (Brocca et al., 2011) or advanced machine learning algorithms (Lohani et al., 2011) to model rainfall and runoff using historical time series data. In addition, physical processes models focus on simulating hydrological processes in a basin based on a set of mathematical equations governing physical processes of water flow and surfaces (Chiew et al., 1993;Birkel et al., 2010;Arnold et al., 1998;Beven et al., 1984;Grimaldi et al., 2013). In general, rainfall–runoff models require relative long term time series data at gauging stations. However, the density

of gauging stations in developing countries is very low and this fact imposes a great obstacle for establishing accurate hydrological models (Fenicia et al., 2008). In addition, large-scale field works and deployments of measuring equipment are necessary for collecting data. Nevertheless, the complex and nonlinear nature of the flood modeling problem also bring about difficulties for hydrological methods and techniques (Sahoo et al., 2006).

In recent year, a new flood modeling approach called "on-off" classification of flood occurrence has been successfully proposed for spatial prediction of flood, also called flood susceptibility (Tien Bui et al., 2016d;Tehrany et al., 2014;Tehrany et al., 2015b). Accordingly, no time series data is required for the model calibration and the establishment of flood models is based on flood inventories (flood class) and non-flood areas (non-flood class). Accordingly, the probability of a pixel in the study area belongs to the flood class is used as flood susceptibility index. Although flood susceptibility map provides no temporal prediction or return period of flood, the flood map is capable delineating highly susceptible areas. Thus, it is a powerful flood analysis tool for decision-makers that could be used in landuse planning and flood management. Literature review shows that data-driven methods integrated with GIS databases have demonstrated their effectiveness and accuracy in large scaled flood susceptible predictions."

**Comment 4: "**Overall, I once again encourage the authors to make substantial improvement both in the methodology framing and writing of manuscript. **"**

**Response**

We agree with the reviewer's comments and have made "substantial improvement both in the methodology framing and writing of manuscript". We believe that the manuscript now has a high quality for the Geoscientific Model Development journal.

**Comments from Anonymous Referee #2**

Corrections:
- Page 1, line 13: distribution of

We have made the correction: replaced "distributionof" by "distribution of"

- Page 5, line 121: by far

We have done this "byfar" to "by far"

- Page 21, line 360: appropriate number of k

We have rewritten the texts "To determine the best k, the EM algorithm that employs Akaike Information Criterion (AIC) (Akaike, 1974) was used"

- Page 27, lines 474 to 476: please make corrections and rewrite.

We have rewritten the texts. We copy the texts in the revised manuscript here for review:

"To confirm the performance of the proposed BayGmmKda model is significantly higher than that of the three benchmark model, the Wilcoxon signed-rank test is employed. The Wilcoxon signed-rank test is widely used to evaluate whether classification outcomes of prediction models are significantly dissimilar (Tien Bui et al., 2016e). Using this test, the *p*-values that obtained from experimental results of the four models can be computed using a threshold value of 0.05. The result of the Wilcoxon signed-rank test is shown in **Table 4**. It is noted that the signs "++", "+", "--", and "-" represent a significant win, a win, a significant loss, and a loss, respectively. The result confirms that the proposed BayGmmKda model achieves significant wins over the other models."

- Page 30, line 521: consideration of the patterns

We have replaced "consideration the patterns" by "consideration of the patterns"
* * *
**Tracked Change Version**

[revised manuscript text omitted]

---

## Author Response (AR3)

Dear **Editor**:

We 'd like to thank the Editor and the Reviewers for the help of reviewing our manuscript. Based on your comments and suggestions, we have revised the manuscript accordingly. The detailed responses of the authors are given in the below section. The tracked change version of the manuscript has also been incorporated into this file.

1) The quality of the writing, especially in the introduction and literature review is not sufficient for publication. The grammatical errors make the manuscript very difficult to read.

**Response to reviewer's comment:**

According to the reviewer' comment, we have revised the paper to improve the quality of writing with the focus on the introduction and literature review. The grammatical errors have been identified and corrected. Please refer to the revised manuscript for the detailed changes.

2) "I simply do not understand your argument regarding missing data and data length. For example, if I were to sample flood locations in a particular year, missing many of the floods and assuming some random locations where I had not sampled were non-flood locations I could use the proposed methods to get a flood susceptibility estimate. I do not see how this can be a probability because you have never seen the vast majority of floods that could occur and you would not know how common the flood you have seen is."

**Response to reviewer's comment:**

We thank the reviewer for kindly reviewing our manuscript. Actually, we are happy to have a well-known flood expert like you to review our work.

Regarding the flood locations (flash flood), in this research, within the Project No B2014-02-21, these flood locations that occurred and effected directly to the infrastructure and people were provide by the Tuong Duong district authority, these floods occurred from 2010-2014 and during tropical rainstorms. Our analysis reveals that these were recurrent flash floods. In other words, these floods occurred in the same locations every year. Some flash floods in mountainous areas were detected based on "flood signals" on Landsat imagery plus fieldworks. This means that almost of all "important" flood locations were used for modeling.

Yes, we have no flood locations before 2010, however, we think that these flood locations before 2010 are almost the same locations of those from 2010-2014. Because with tropical rainstorms for five years 2010-2014, almost flash flood locations were revealed.

We understand the concern of the reviewer on non-flood locations may be randomly sampled in "flood areas" if these flood areas, and if this is the case, the model will get problem.

However, this is not the case in this study area. This is because non-flood locations were sampled in *these mountain ridges* of the study area. This was carried out using the DEM of the study area and GIS techniques. More specifically, these mountain ridges in the study area were derived using DEM and the toposhape module in Idrisi Selva software. Because no flood occurred in these mountain ridges, the non-flood locations in this study is valid and do not located within any flood region.

I hope the above explanations are very clear to your concern.

"Alternatively, I could have gathered data over a 100 year period on every flood in the area and used those data in your model. Under this scenario I would have many more flooded pixels and presumably a higher flood susceptibility, unless the method accounts for data record length and the potential for missing data.

**Response to reviewer's comment:**

Although we agree with this comment, however, we would like to explain as follows:

- We have no data on 100 year periods for the study area. More importantly, these flood locations were not from gauge stations, therefore, it is very difficult to such data, even in developed countries.

- This flash floods modeling approach based on a statistical assumption that flash floods will occur in the future under the same geo-environmental conditions that produced them in the past. Therefore, if we used flood locations i.e. 50 years, or 100 years ago, the geo-environmental conditions 50 year ago may very different compared to the present time. In other words, some flash flood locations in the past "i.e. 50 year ago" may not be flood now due to the fast developments (i.e. new hydro-powers constructed in the study area) as well as deforestation or new forests. Consequently, using of very old flood locations may be bias for this case study.

Based on the above explanations, we think that these are very clear to you.

"I'm not saying your method lacks merit but it needs to be clear how it differs, and in what respect it might be inferior, to frequency based methods that attempt to estimate the probability of a location flooding in a given year."

**Response to reviewer's comment:**

Our approach is totally different compared to those from traditional hydrologic modeling method. This has been clearly presented in the revised manuscript as we use a new way called "on-off" classification approach.

The resulting map from our model does not provide "the probability of a location flooding in a given year". The resulting map provides only spatial probability of flood also called flood susceptibility, no temporal dimension is included. In other words, the flood susceptibility map will delineate "high and very high susceptibility flood" that is useful for land use planning and also for further larger scale flood study.

We think that these explanations are very clear to you.

Once again, we thank the reviewer for spending your expensive time and expertise to comment on our manuscript. Based on your comments, it is clear that the quality of the manuscript has been significantly improved. We believe that our approach is a meaningful contribution to literature.
* * *

[revised manuscript text omitted]

---

## Author Response (AR4)

Dear **Editor**:

We 'd like to thank you for the help of reviewing our manuscript. Based on your comments and suggestions, we have revised the manuscript accordingly.

The detailed responses of the authors are given in the below section. The tracked change version of the manuscript has also been incorporated into this file.

Comments to the Author:

Dear Tien Bui Dieu,

Thank you for your response to my previous question. I believe I know understand your method and I would like to recommend your article for publication. I have consulted with the journals copy editors on the standard of English because it is not yet of publication standard. However they have indicated to me that they will be able to work on the manuscript prior to publication.

**Reply to reviewer's comment**: Thanks for your recommendation. This is a great motivation for us to perform better research work in the future.

I'm sorry to suggest more changes but I my final read through I spotted 11 minor amendments you will want to consider. I also think the justification of the data length you have provided in the response to my question should be written formal into the manuscript.

**Reply to reviewer's comment**: According to your comment, we have provided that justification of the data length on line 158 page 6 as follows:

"Although the data for this study is collected from 2010 to 2014, they were recurrent flash floods which occurred during tropical typhoons. Thus, it is reasonable to conclude that all significant flash flood locations in the study area have been revealed and determined. It should be noted that due to the statistical assumption used in this study, the inclusion of flood locations in the far past (i.e. before the year of 2009) for flood susceptibility analysis may cause bias. It is because the construction of new hydropower dams such as Ban Ve (from 2010) and Nam Non (from 2011) and deforestation/forestation have changed the geo-environmental conditions in the study area (Dao, 2017;Manley et al., 2013). In other words, the geo-environmental conditions of the far past are very different to those of the present time; therefore, flood locations in the far past should not be included in the current analysis." Specific corrections

1) Line 15: "the posterior probability of flood, which is the output of the BayGmmKda model, is used as flood susceptibility index" should read "the posterior probability output of the BayGmmKda model, is used as flood susceptibility index" as it is not the probability of flood due to the lack of a frequency component.

**Reply to reviewer's comment**: We have revised the sentence as follows:

As a result, the posterior probabilistic output of the BayGmmKda model is used as flood susceptibility index.

2) Line 31: The sentence starting "Particularly, Vietnam is a storm centre" does not make sense, please reword

**Reply to reviewer's comment**: We have revised the sentence as follows:

Located in this region, Vietnam is a storm center at the Western Pacific and this nation has faced the destructive consequence of flooding in many of its provinces.

3) Line 38: "flood forecasting" technically flood forecasting is wrong in this context you should be talking about predicting flood hazard/risk for land use planning

**Reply to reviewer's comment**: We have revised the sentence as follows:

Hence, an accurate model for evaluating flood hazard for land use planning becomes a crucial need for land-use planning as well as establishment of disaster mitigation strategies.

4) Line 50: "enhancing the prediction accuracy" relative to what

**Reply to reviewer's comment**: We have revised the sentence as follows:

Given these motivations, this study proposes a novel methodology designed for achieving a high prediction accuracy as well as deriving probabilistic evaluations of flood susceptibility in a regional scale.

5) Line 54: "To yield probabilistic outputs of flood," should be "To yield probabilistic outputs of flood susceptibility,"

**Reply to reviewer's comment**: We have revised the sentence as follows:

To yield probabilistic outputs of flood susceptibility, this study proposes a Bayesian framework established on the basis of an integration of Gaussian mixture model (GMM) and the Kernel Fisher Discriminant Analysis (KFDA).

6) Line 72-73: "relatively simple to more sophisticated methodologies" could you give an example of what you mean by relatively simple and sophisticated because its not clear what you mean

**Reply to reviewer's comment**: We have revised the sentence to avoid confusion of the readers as follows:

Following this trend, various flood analyzing tools have been developed (Winsemius et al., 2013;Papaioannou et al., 2015;Gao et al., 2017;Alfieri et al., 2014).

7) Line 77: "providing flood predictions" should be "providing discharge predictions". You will then need a hydrodynamic model in the case of the statistical models and hydrological models to estimate flooding.

**Reply to reviewer's comment**: We have revised the sentence as follows:

Thus, these models are capable of providing discharge predictions both in space and time.

8) Line 87: The Neal et al paper doesn't use a hydrological model. It's a statistical event generation technique based on river gauge data and probably belongs in the earlier statistical section

**Reply to reviewer's comment**: We have relocate the mentioned reference as follows:

Various types of rainfall–runoff models can be found in literature, varying from empirical models to highly sophisticated physical processes. Empirical models could be established based on statistical techniques (Brocca et al., 2011;Neal et al., 2013) or advanced machine learning algorithms (Lohani et al., 2011); such models can be effectively employed to analyze rainfall and runoff on the basis of historical time series data. In addition, physical processes models focus on simulating hydrological processes in a basin based on a set of mathematical equations governing physical processes of water flow and surfaces (Aronica et al., 2012;Chiew et al., 1993;Beven et al., 1984;Birkel et al., 2010;Grimaldi et al., 2013).

9) Line 91: For me this sentence adds nothing and should be removed or used differently "Nevertheless, the complex and nonlinear nature of the flood modeling problem also brings about difficulties for the hydrological methods and techniques (Sahoo et al., 2006)."

**Reply to reviewer's comment**: We have removed this sentence to avoid confusion of the readers.

10) Line 94: "susceptibility prediction of flood" previously you have used the wording flood susceptibility index. Could you use the same definition for your output throughout the manuscript

**Reply to reviewer's comment**: We have changed the term as your suggestion.

11) Line 99: Somewhere here you need to say that the results depend on having sufficient training data

**Reply to reviewer's comment**: We have added the statement you recommended as follows:

[revised manuscript text omitted]